



# Source Apportionment of Carbonaceous Aerosols in Beijing with Radiocarbon and Organic Tracers: Insight into the Differences between Urban and Rural Sites

Siqi Hou[1*], Di Liu[1,2], Jingsha Xu[1], Tuan V. Vu[1,3], Xuefang Wu[4], Deepchandra Srivastava[1], Pingqing Fu[5], Linjie Li[6,7], Yele Sun[6], Athanasia Vlachou[8], Vaios Moschos[8], Gary Salazar[9], Sönke Szidat[9], André S.H. Prévôt[8], Roy M. Harrison[1], Zongbo Shi[1]

[1] School of Geography Earth and Environmental Science, University of Birmingham, Birmingham, B15 2TT, UK
[2] Now at: Institute of Atmospheric Physics, Chinese Academy of Sciences, Beijing, 100029, China
[3] Now at: School of Public Health, Imperial College London, London, UK
[4] School of Earth Sciences and Resources, China University of Geosciences, Xueyuan Road 29, 100083, Beijing, China
[5] Institute of Surface-Earth System Science, Tianjin University, Tianjin 300072, China
[6] State Key Laboratory of Atmospheric Boundary Layer Physics and Atmospheric Chemistry Institute of Atmospheric, Physics, Chinese Academy of Sciences, Beijing, 100029, China
[7] now at: Department of Chemistry and Molecular Biology, University of Gothenburg, Gothenburg 41296, Sweden
[8] Laboratory of Atmospheric Chemistry, Paul Scherrer Institute, Villigen PSI, CH-5232, Switzerland
[9] Department of Chemistry and Biochemistry & Oeschger Centre for Climate Change Research, University of Bern, Bern, CH-3012, Switzerland

*Siqi Hou and Di Liu made equal contributions.

*Correspondence to*: Roy M. Harrison (r.m.harrison@bham.ac.uk), Zongbo Shi (z.shi@bham.ac.uk)

**Abstract.** Carbonaceous aerosol is the dominant component of fine particles in Beijing. However, it is challenging to apportion its sources. Here, we applied a newly developed method which combined radiocarbon ($^{14}$C) with organic tracers to apportion the sources of fine carbonaceous particles at an urban (IAP) and a rural (PG) site of Beijing. PM$_{2.5}$ filter samples (24-h) were collected at both sites from 10 November to 11 December 2016 and from 22 May to 24 June 2017. $^{14}$C was determined in 25 aerosol samples (13 at IAP and 12 at PG) representing low pollution to haze conditions. Biomass burning tracers (levoglucosan, mannosan and galactosan) in the samples were also determined using GC-MS. Higher contributions of fossil-derived OC (OC$_f$) were found at the urban site. OC$_f$ to OC ratio decreased in the summer samples (IAP: $67.8 \pm 4.0\%$ in winter and $54.2 \pm 11.7\%$ in summer; PG: $59.3 \pm 5.7\%$ in winter and $50.0 \pm 9.0\%$ in summer) due to less consumption of coal in the warm season. A novel extended Gelencsér method incorporating the $^{14}$C and organic tracer data was developed to estimate the fossil and non-fossil sources of primary and secondary OC (POC and SOC). It showed that fossil-derived POC was the largest contributor to OC ($35.8 \pm 10.5\%$ and $34.1 \pm 8.7\%$ in winter time for IAP and PG, $28.9 \pm 7.4\%$ and $28.9 \pm 9.6\%$ in summer), regardless of season. SOC contributed $50.0 \pm 12.3\%$ and $47.2 \pm 15.5\%$ at IAP, and $42.0 \pm 11.7\%$ and $43.0 \pm$





13.4% at PG in the winter and summer sampling periods respectively, within which the fossil-derived SOC was predominant and contributed more in winter. The non-fossil fractions of SOC increased in summer due to a larger biogenic component.

Concentrations of biomass burning OC (OC$_{bb}$) are resolved by the extended Gelencsér method with average contributions (to total OC) of 10.6 ± 1.7% and 10.4 ± 1.5% in winter at IAP and PG, and 6.5 ± 5.2% and 17.9 ± 3.5% in summer, respectively. Correlations of water-insoluble OC (WINSOC), water-soluble OC (WSOC) with POC and SOC showed that although WINSOC was the major contributor to POC, a non-negligible fraction of WINSOC was found in SOC for both fossil and non-fossil sources especially during winter. In summer, a greater proportion of WSOC from non-fossil sources was found in

SOC. Comparisons of the source apportionment results with those obtained from a Chemical Mass Balance model were generally good, except for the cooking aerosol.

## 1 Introduction

Carbonaceous aerosols, often one of the most abundant components (20-80%) in atmospheric aerosol particles, have a crucial impact on the global climate, air quality and human health (He et al., 2001; Huang et al., 2014; Jimenez et al., 2009;

Song et al., 2007; Zhou et al., 2018). The total content of carbonaceous aerosols (i.e., total carbon, TC) can be divided into organic carbon (OC) and elemental carbon (EC) according to their physical, chemical and optical properties. The source of EC is from incomplete combustion of fossil fuel or biomass, while OC mainly originates from primary emissions from sources such as coal combustion, traffic emissions, cooking, and biomass burning as well as from gas-particle conversion (Yang et al., 2016). It is very challenging to quantify the contributions from different sources to OC and EC because of the

limited information on the sources, atmospheric loading and composition of organic aerosols (Huang et al., 2014). Radiocarbon ($^{14}$C) analysis is a powerful tool for the quantification of fossil and non-fossil contributions to carbonaceous aerosols, as non-fossil sources contain a high contemporary $^{14}$C content, while the fossil fractions are free of $^{14}$C (Zotter et al., 2014; Bernardoni et al., 2013; Liu et al., 2013; Szidat et al., 2004; Szidat et al., 2006; Szidat et al., 2009). A previous study in north-east China found a dominant fossil-fuel contribution to EC (76 ± 11 %), and that non-fossil sources are major

contributors to OC (66 ± 11 %) (Zhang et al., 2016). Non-fossil sources of OC were major contributors to the fine particle pollution in Beijing during the APEC summit (Liu et al., 2016a). Moreover, clear seasonal trends of non-fossil and fossil source contributions to water-insoluble OC (WINSOC) and water-soluble OC (WSOC) were found. Non-fossil sources were the major contributor (59 %) to WINSOC in summer and autumn, whereas fossil fuel emissions were predominant in winter and spring (Liu et al., 2013). Proportions of non-fossil sources in TC and WSOC associated with biogenic emissions

increased during spring and summer with maxima (85 % and 117 %, respectively) in May (Pavuluri et al., 2013). However, $^{14}$C measurements do not permit direct discrimination of specific sources (e.g. biomass burning or secondary OC - SOC) of modern carbon. A combination with other techniques gives further insight into the characteristics of SOC (Minguillon et al.,



2011; Szidat et al., 2006; Szidat et al., 2009; Yttri et al., 2011). For example, applying Latin hypercube sampling with different OC/EC ratios, relative contributions of primary and secondary organic carbon were estimated (Zhang et al., 2016).

[14]C analysis combined with AMS-PMF data has contributed to the identification of sources from primary emissions and secondary formation (Barrett et al., 2015; Zhang et al., 2018; Zhang et al., 2017; Vlachou et al., 2018).

The Gelencsér method provides a first-order source apportionment of organic aerosol from fossil fuel combustion, biomass burning, biogenic emissions and secondary organic aerosol, using measurements of specific organic tracers emitted by fossil and non-fossil sources and their OC/EC ratios derived from literature (Gelencsér et al., 2007). This method was derived in a

European context, but for China the inclusion of food cooking and coal combustion is required. Based upon this consideration, an extended Gelencsér method that includes quantification of fossil and contemporary EC and OC by [14]C analysis has been developed in this study. The diversity of fuel types and combustion conditions make the selection of OC/EC ratios for biomass burning difficult due to large uncertainties. For non-fossil sources of SOC, quantification by the method of Gelencsér et al. (2007) is totally dependent on the source apportionment of OC from biomass burning, and thus a

cautious selection of ratios has been adopted in this study.

Biomass burning is an important source of both EC and OC, which can affect large areas of the world through long-range transport (Andreae and Merlet, 2001). It is also a key component when applying source apportionment by the Gelencsér method. As levoglucosan (1,6-anhydro-β-D-glucopyranose, LG) is an almost specific biomass burning tracer as the main pyrolysis product from cellulose (Puxbaum et al., 2007; Simoneit et al., 1999), concentrations of OC from biomass burning

can be obtained by multiplying LG with suitable OC/LG ratios (Gelencsér et al., 2007; Zdrahal et al., 2002). However, the wide range of OC/LG ratios associated with changes in the biofuel types and combustion conditions cause great uncertainty in the estimation (Cheng et al., 2013; Fu et al., 2012; Gelencsér et al., 2007). To mitigate these differences from types of material and the burning conditions, a typical ratio of 12.2–12.5 (12.3 on average) was documented in Andreae and Merlet (2001) by considering the biofuels of savanna and crop residues. This ratio is widely accepted in many studies (Andreae and

Merlet, 2001; Fu et al., 2012; Zhang et al., 2008a; Zhang et al., 2007), and has been used to estimate the contributions of biomass burning in Beijing which ranged from 8–50 % (Cheng et al., 2013; Kang et al., 2018; Liu et al., 2017; Zhang et al., 2008a; Zhang et al., 2017). However, a single ratio is not representative of all local conditions, and more specific methods are needed. [14]C analysis can provide accurate concentrations of EC from biomass burning, assuming that EC from non-fossil sources is exclusively from biomass burning. Introducing $EC_{nf}$ concentrations from [14]C analysis into the Gelencsér method

of using OC/LG ratios provides valuable extra information, and allows the method to be extended to include other sources.

Beijing, capital of China, has experienced severe $PM_{2.5}$ pollution for decades. This has been subjected to extensive research. However, few studies have sought to differentiate the fossil and non-fossil sources of SOC, even though they provide key



information on the precursors and formation mechanisms of SOC. In this study, measurements of PM$_{2.5}$, OC and EC along with biomass burning tracers were conducted simultaneously at urban and rural sites of Beijing in the winter of 2016 and
summer of 2017. $^{14}$C measurements of EC, OC, WINSOC and WSOC were carried out subsequently on filter samples to enable source apportionment of fossil vs non-fossil sources. A novel extended Gelencsér method combining $^{14}$C analysis has been developed to explore the source apportionment of OC and EC, with SOC from fossil and non-fossil sources being quantified. The source apportionment results were compared with those by Chemical Mass Balance (CMB). Correlations of WINSOC and WSOC with different sources of OC were also performed to study the formation mechanisms of SOC.

## 2 Methodology

### 2.1 Aerosol Sampling

Daily PM$_{2.5}$ (particles with aerodynamic diameter less than 2.5 μm) samples were collected at an urban site (39.98° N, 116.39° E, Institute of Atmospheric Physics, IAP), and a rural site (40.17° N, 117.05° E, Pinggu, PG) in Beijing during a winter campaign (10 November–11 December 2016) and a summer campaign (22 May–24 June 2017) as part of the
Atmospheric Pollution and Human Health in a Chinese megacity (APHH-China) programme; further information on the sampling sites is available in Shi et al. (2019). The urban site is a typical urban background site but may be subject to multiple local influences such as cooking emissions from nearby restaurants. The rural site is located in Pinggu District, close to a village surrounded by farmland. It is ~60 km northeast from urban Beijing at the junction of Beijing, Tianjin and Hebei provinces. A two-lane road is about 200–300 meters north of the sampling site, but its traffic volume is relatively low.

Hi-Vol air samplers (Tisch, USA) with a flow rate of 1.1 m$^3$ min$^{-1}$ were collected on pre-combusted (450 °C, 6 h) quartz filters (Pallflex, 8×10 inch). Field blanks were collected by placing filters onto the filter holder for a few minutes without pumping before and after the campaign. After sampling, each exposed or blank filter was wrapped individually with aluminium foil and stored at -20 °C in the dark prior to analysis. The details of the sample collection are described elsewhere (Shi et al., 2019).

### 2.2 Chemical Analysis

#### 2.2.1 OC, EC and major inorganic ions

OC and EC mass concentrations were determined with the DRI2015 carbon analyser with the EUSAAR_2 (European Supersites for Atmospheric Aerosol Research) transmittance protocol. Replicate analyses were conducted once every ten samples. Blank samples (corresponding to 0.40 and 0.01 μg m$^{-3}$ for OC and EC) were analysed to correct the sample results.
The limits of detection of OC and EC were estimated to be 0.03 and 0.05 μg m$^{-3}$. Details of the OC/EC measurement method



are described elsewhere (Paraskevopoulou et al., 2014). Major ions including $SO_4^{2-}$, $NO_3^-$, $NH_4^+$, $Na^+$, $K^+$ and $Cl^-$ were determined on water extracts using an ion chromatograph (Dionex, Sunnyvale, CA, USA), with detection limits less than 0.01 µg m$^{-3}$. The uncertainties for OC and EC were less than 10% and less than 5% for inorganic ions (Xu et al., 2020a).

### 2.2.2 Biomass Burning Tracers

The methodology to determine biomass burning tracers, including levoglucosan, mannosan and galactosan was described elsewhere (Fu et al., 2016). Recoveries for target compounds were better than 80 % as obtained by spiking standards to pre-combusted quartz filters followed by extraction and derivatization. Field blank filters were analysed by the procedure used for the samples above, but no target compounds were detected. Duplicate analyses showed analytical errors less than 15 %.

### 2.3 Radiocarbon ($^{14}$C) Analysis

The $^{14}$C in total carbon (TC), water-insoluble TC (WINSTC) and EC was determined on 25 (13 from IAP and 12 from PG) time-integrated Hi-Vol PM$_{2.5}$ quartz fibre (QF) filter samples. Samples collected during both haze and non-haze days were selected in winter to better understand the pollution sources. PM$_{2.5}$ concentrations on 22 November and 1 December at IAP and PG sites were lower than 75 µg m$^{-3}$ and regarded as non-haze air days, in contrast to other wintertime samples collected during haze pollution days. During the summer, typical samples were selected with PM$_{2.5}$ concentrations of 42.5 ± 26.5 and 135 42.7 ± 21.2 µg m$^{-3}$ at IAP and PG, respectively. The concentrations of PM$_{2.5}$, EC, OC and the corresponding non-fossil fractions of these selected days are shown in Table S1.

The method of $^{14}$C measurement of carbonaceous aerosols has been described elsewhere (Agrios et al., 2015; Levin et al., 2010; Szidat et al., 2014; Vlachou et al., 2018; Zhang et al., 2012; Zhang et al., 2016; Zotter et al., 2014). The $^{14}$C of TC and WINSTC was measured by using an one-step protocol under pure O$_2$ (99.9995%) at 760 ℃ for 400 s (Vlachou et al., 2018) 140 using an Elemental Analyzer coupled with the accelerator mass spectrometer Mini Carbon Dating System (MICADAS) at the Laboratory for the Analysis of Radiocarbon (LARA, University of Bern) (Zhang et al., 2012; Szidat et al., 2014). The EC fraction was separated by an OC/EC analyser (Model 4L, Sunset Laboratory, USA) with the use of the Swiss_4S protocol (Zhang et al., 2012), which was coupled online with the MICADAS (Agrios et al., 2015). Each filter sample was extracted with water before the measurements to minimize the charring effect during the separation of EC from the WINSOC.

$^{14}$C results were expressed as fractions of modern (f$_M$), i.e., the fraction of the $^{14}$C/$^{12}$C ratio of the sample related to that of the reference year 1950. The data analysis was carried out accounting for the blank correction (one field blank per site was analysed, not relevant for EC), decay of $^{14}$C since the 1950's, nuclear bomb correction, charring of WINSOC (~1%) and EC yield after OC removal (IAP: 62± 6%; PG: 76 ± 8%) (Zhang et al., 2012; Zhang et al., 2016; Zotter et al., 2014).





Non-fossil fractions ($f_{NF}$) were determined from their corresponding $f_M$ values and reference values for pure non-fossil
sources by:

$$f_{NF} = \frac{f_M}{f_{NF,ref}} \qquad (1)$$

Different values for the $f_{NF,ref}$ were applied for the bomb peak correction (Levin, et al. 2010). For EC, the $f_M$ is $1.10 \pm 0.05$
(Lewis et al., 2004; Palstra and Meijer, 2014), given that biomass burning is assumed to be the only non-fossil source of EC.
For OC, it is calculated as

$$f_{NF,ref} = p_{bio} \times f_{M,bio} + p_{bb} \times f_{M,bb} \qquad (2)$$

where $f_{M,bb}$ and $f_{M,bio}$ is coming from biomass burning and biogenic sources respectively, which are $1.10 \pm 0.05$ and $1.023 \pm 0.015$ (Lewis et al., 2004; Zotter et al., 2014), while $p_{bio}$ and $p_{bb}$ are the proportions of biogenic source and biomass burning
respectively, which are 0.9 and 0.1 in winter and 0.5 and 0.5 in summer (Levin et al., 2010).

Analogously, the non-fossil fractions of OC, WSOC and WINSOC ($f_{NF,OC}$, $f_{NF,WSOC}$ and $f_{NF,WINSOC}$) were calculated by
following a mass balance-like approach:

$$OC_{nf} = OC \times f_{NF,OC} = TC \times f_{NF,TC} - EC \times f_{NF,EC} \qquad (3)$$

$$WSOC_{nf} = WSOC \times f_{NF,WSOC} \approx WSOC \times f_{NF,WSTC} = TC \times f_{NF,TC} - WINSTC \times f_{NF,WINSTC} \qquad (4)$$

$$WINSOC_{nf} = WINSOC \times f_{NF,WINSOC} = OC \times f_{NF,OC} - WSOC \times f_{NF,WSOC} \qquad (5)$$

where TC and EC are the concentrations of total and elemental carbon, respectively, and $f_{NF,TC}$, $f_{NF,EC}$, $f_{NF,WINSTC}$ are the non-
fossil fractions of TC, EC and WINSTC, respectively. The fraction of fossil-fuel sources was calculated by $f_{FF} = 1 - f_{NF}$. The
uncertainties were determined by error propagation. The mass concentration errors were assumed to be 10 % for EC and 6 %
for OC and TC (typical values for EUSAAR2) (Zhang et al., 2016).

## 2.4 Extended Gelencsér method including [14]C data

An extended Gelencsér method including [14]C data was developed to quantify the fossil and non-fossil sources of primary and
secondary OC (POC and SOC) along with OC from biomass burning and cooking ($OC_{bb}$ and $OC_{ck}$). The equations for the
extended Gelencsér method are list in Table 1. The detailed selection of the OC/EC ratios will be discussed in Sect. 3.2.



## 2.5 Chemical Mass Balance (CMB) model and AMS/ACSM-PMF Analysis

Results on the same sets of samples from a chemical mass balance (US EPA CMB8.2) model and AMS/ACSM-PMF analysis (Positive matrix factorization analysis of data from online Aerodyne Aerosol Mass Spectrometer at IAP and Aerosol

Chemical Speciation Monitor at PG) were compared with [14]C-based source apportionment. Details on the experimental details and data analyses can be found in Xu et al. (2020b) and Wu et al. (2020). The CMB utilizes a linear least squares solution considering both uncertainties in source profiles and ambient measurements to ensure reliable fitting results. In order to better represent the source characteristics, the source profiles applied in this model were mostly from local studies in China (Cai et al., 2017; Wang et al., 2009; Zhang et al., 2007; Zhang et al., 2008b; Zhao et al., 2015), except vegetative

detritus (Rogge et al., 1993; Wang et al., 2009). The detail of selecting organic marker species can be found in Yin et al. (2010; 2015).

Experimental details of the AMS and ACSM-PMF method can be found elsewhere (Ng et al., 2011; Sun et al., 2016; Xu et al., 2019). The ACSM data were analysed for the mass concentrations and size distributions of non-refractory submicron aerosol (NR-PM$_1$) species using the high-resolution data analysis software package PIKA (Sun et al., 2020). Positive matrix

factorization was performed on high-resolution mass spectra of V-mode and W-mode to retrieve potential OA factors from different sources (Paatero and Tapper, 1994; Ulbrich et al., 2009). The OM/OC factor used for cooking OA (COA) is 1.38 (Xu et al., 2019).

## 3 Results and discussion

### 3.1 Overall results

#### 3.1.1 Characteristics of PM$_{2.5}$, OC and EC concentrations

Mass concentrations of PM$_{2.5}$, OC, EC and biomass burning tracers are shown in Fig. 1 and are summarized in Table 2 along with the meteorological conditions during the observation campaign in IAP and PG. The average concentrations of PM$_{2.5}$ were $91.2 \pm 63.7$ µg m$^{-3}$ and $99.7 \pm 77.8$ µg m$^{-3}$ at the IAP and PG sites respectively in winter, and $30.2 \pm 14.8$ µg m$^{-3}$, and $27.5 \pm 12.9$ µg m$^{-3}$ at the IAP and PG sites in summer. The highest 24-h concentration in winter is 239.9 µg m$^{-3}$ (IAP) and

294.3 µg m$^{-3}$ (PG), and more than 53 % and 46 % of the days have a daily PM$_{2.5}$ concentrations higher than the Chinese air quality standard (PM$_{2.5}$ concentrations exceeding 75 µg m$^{-3}$ are defined as haze conditions) in IAP and PG during the observation period. In summer, the air quality was improved, with PM$_{2.5}$ concentrations ranging from 12.2 to 78.8 µg m$^{-3}$ and 11.6 to 70.3 µg m$^{-3}$ in IAP and PG, respectively.

Organic carbon and elemental carbon (OC and EC) are important constituents of PM$_{2.5}$, accounting for $30.9 \pm 9.3$ % and 43.6

$\pm 17.9$ % of PM$_{2.5}$ mass at the IAP and PG sites in winter, and $26.8 \pm 9.2$ % and $37.3 \pm 12.6$ % in summer. The





concentrations of EC showed a strong correlation with OC at both sites during the winter and summer (Table 3). The average EC concentrations for $^{14}$C analysis varied from $3.8 \pm 2.1$ µg m$^{-3}$ (IAP) and $5.4 \pm 2.6$ µg m$^{-3}$ (PG) in winter and $1.1 \pm 0.3$ µg m$^{-3}$ (IAP) and $2.0 \pm 0.7$ µg m$^{-3}$ (PG) in summer for the urban and rural sites respectively. The mass concentration of OC for $^{14}$C analysis was 4.1–44.9 µg m$^{-3}$ and 12.1–85.0 µg m$^{-3}$ at IAP and PG in winter, and 4.7–12.7 µg m$^{-3}$ and 6.2–17.9 µg m$^{-3}$ in

summer. The selected samples are well representatives as their concentrations were very close to those from the whole campaign. The average OC concentration for $^{14}$C in winter was 3.2 and 4.3 times higher than in summer at IAP and PG, respectively. The OC/EC ratios at IAP and PG were in the range of 4.1–14.9 and 6.2–14.6 in winter, and 4.6–14.8 and 4.4–28.3 in summer, which were all higher than 2.0 or 1.1 (Chow et al., 1996; Castro et al., 1999), suggesting an important contribution from secondary organic carbon (SOC).

### 3.1.2 Fossil and non-fossil sources of EC and OC based on radiocarbon ($^{14}$C) analysis

Fig. 2 shows the absolute concentrations of fossil and non-fossil fractions of OC and EC, and the relative contributions. EC$_f$ refers to EC from coal combustion and liquid fossil fuel (i.e., mainly vehicle emissions) and EC$_{nf}$ from biomass burning (Gray and Cass 1998). As shown in Fig. 2, high concentrations of EC$_f$ and EC$_{nf}$ were found in both urban and rural sites in winter, suggesting elevated emissions from primary fossil and non-fossil sources like coal combustion and biomass burning.

Residential coal consumption and biomass burning are still important in winter especially in rural areas, due to intensive heating activities in the cold season. EC$_f$ contributed $7.6 \pm 2.1$ % and $6.0 \pm 1.4$ % to TC at the IAP and PG sites in winter and $6.9 \pm 1.6$ % and $8.9 \pm 2.6$ % in summer. Most of the fossil fractions of EC are within the range of previous studies in urban Beijing (67–96% of EC) (Liu et al., 2020; Liu et al., 2017; Zhang et al., 2016; Zhang et al., 2017). Higher contributions of EC$_f$ were found on the polluted days in wintertime (Table S1), showing that the PM$_{2.5}$ concentrations may be elevated due to

direct emission from coal combustion and vehicle exhaust. Fractions of EC$_{nf}$ increased slightly in summer due to the regional open burning activities during the post-harvest period of wheat, which is common in Northern China (Li et al., 2016; Yan et al., 2006).

OC from fossil sources (OC$_f$) arises mainly from coal combustion and vehicle emissions, while OC from non-fossil sources (OC$_{nf}$) comes mainly from biomass burning, biogenic emissions and cooking. Although fossil sources are the main

contributor to OC ($67.8 \pm 4.0$ % and $59.3 \pm 5.7$ % at IAP and PG sites in winter, compared to $54.2 \pm 11.7$ % and $50.0 \pm 9.0$ % in summer), relative contributions of OC$_{nf}$ were significantly increased in summer. These results are similar to a previous study in urban Beijing, which found a contribution of $66 \pm 11$ % ranging from 45 to 82 % in PM$_1$ collected in 2013–2014 (Zhang et al., 2017). The increased contribution of OC$_{nf}$ in summer is likely to due to the enhancement of biomass burning and biogenic emissions (both primary and through SOC) as well as the decline in emissions from fossil sources such as coal

burning, considering that cooking OC emission is unlikely to change with season.



Fossil and non-fossil sources of WSOC and WINSOC were also quantified by the $^{14}$C measurement. Among the four fractions, WINSOC$_f$ had the highest contribution to TC in winter (37.9 ± 4.5 % at IAP and 36.2 ± 4.7 % at PG) followed by WSOC$_f$ (22.1 ± 5.2 %) at IAP and WINSOC$_{nf}$ (20.8 ± 3.4 %) at PG. In summer, the fraction of WINSOC$_f$ fell to 24.4 ± 9.3 % at IAP and 21.8 ± 4.6% at PG, accompanied by increased fossil and non-fossil contributions from WSOC fractions. The

increase of WSOC$_f$ fractions in summer implied an enhanced contribution from oxidised VOCs and aged primary fossil-derived OC, and WSOC$_{nf}$ is probably associated with biomass burning and secondary OC. Moreover, WINSOC was dominated by fossil sources at both sites in winter (IAP: 72.6 ± 3.6 %; PG: 63.4 ± 6.5 %), while the non-fossil fractions significantly increased from winter to summer. Similarly, WSOC was mainly fossil-derived in winter, while it tends to be accumulated in non-fossil fractions in summer (fossil fraction in WSOC, IAP: 60.5 ± 6.6 % in winter, 50.8 ± 12.3 % in

summer; PG: 51.6 ± 8.7 % in winter, 47.7 ± 17.4 % in summer). The contribution of OC$_{nf}$ to OC increased with both non-fossil fractions in WINSOC and WSOC increasing. Even though WINSOC$_{nf}$ and WSOC$_{nf}$ cannot be attributed specifically to biomass burning, biogenic emissions, cooking or secondary formation, it is likely that biogenic- derived POC and SOC make a pronounced contribution. Details of contributions of each fraction to primary and secondary OC will be discussed in Sect. 3.2.

### 3.2 Source apportionment by an extended Gelencsér method


Gelencsér et al. (2007) reported a method for the source apportionment of carbonaceous aerosol into fractions from biomass burning, road traffic and secondary organic aerosol, applicable to Europe where these are the dominant sources. In order to use the same methodological concepts in China, the method required extending to include two further sources: coal combustion and cooking. To do so, the $^{14}$C data were utilized.

### 3.2.1 Biomass burning


Levoglucosan (LG) is a typical biomass burning tracer, as the main pyrolysis product from cellulose. Much higher concentrations of LG were observed in the winter (311 ± 193 ng m$^{-3}$ at IAP and 634 ± 483 ng m$^{-3}$ at PG) than those in the summer (27.9 ± 29.6 ng m$^{-3}$ at IAP and 74.0 ± 34.2 ng m$^{-3}$ at PG). In addition, LG concentrations at the rural site were higher than those at the urban site in both winter and summer. This pattern is consistent with previous measurements in Table 4

(Chen et al., 2018; Kang et al., 2018; Li et al., 2018; Liu et al., 2016b; Salma et al., 2017; Sullivan et al., 2019; Yan et al., 2019; Zhu et al., 2017). The Pearson correlations of LG with PM$_{2.5}$, OC and EC at IAP and PG are shown in Table 3. During winter, LG correlated well with PM$_{2.5}$, OC and EC at PG with correlation coefficients of 0.89, 0.89 and 0.81 respectively. These are higher than those at IAP (correlation coefficients of 0.56, 0.60 and 0.74, respectively), suggesting a more significant influence of biomass burning upon PM$_{2.5}$ in PG. The correlation coefficients in summer were much lower than

those in winter for both sites, evidencing a reduced contribution of biomass burning activities to PM$_{2.5}$. During the





wintertime, the increasing use of biofuel for heating exacerbates the biomass pollution, and the stable atmospheric conditions also enhance the accumulation of LG (Shi et al., 2019). Compared with widely used cleaner fossil energy (i.e., natural gas, electricity, and liquefied petroleum gas) and renewable energy (i.e., solar energy) available in urban areas, the rural households are still largely using straw and wood for cooking and heating (Hou et al., 2017). Moreover, open burning of

crop residues during the post-harvest months (May to July and October to November in North China) in rural areas is still frequently performed in spite of prohibition by the government (Chen et al., 2017; Li et al., 2016). Water-soluble potassium ion ($K^+$) has been used as biomass burning tracer previously, due to its good relationship with LG. However, in this study, the Pearson correlation coefficients of $K^+$ with LG are 0.51 and 0.86 in winter and 0.85 and 0.51 in summer for IAP and PG (Table 3), indicating other non-biomass sources of $K^+$. Indeed, the sources of $K^+$ in the atmosphere are diverse, including sea

salt, cooking, dust, coal combustion, and waste incineration, which makes $K^+$ less suitable as biomass burning tracer (Zhang, et al., 2010).

According to Gelencsér et al. (2007), EC from biomass burning ($EC_{bb}$) can be derived by multiplying the LG concentrations first with $(OC/LG)_{bb}$ to give $OC_{bb}$ and then with $(EC/OC)_{bb}$. However, the $(OC/LG)_{bb}$ ratio is highly variable depending on the type of material and the conditions of burning, and the ratios can vary by orders of magnitude. Therefore, EC

concentrations from non-fossil sources determined by radiocarbon analysis ($EC_{nf}$), assumed to arise almost solely from biomass burning were used as an estimate of $EC_{bb}$. Hence,

$$EC_{nf} \approx EC_{bb} = LG \times (OC/LG)_{bb} \times (EC/OC)_{bb} \qquad (6)$$

The ratios of levoglucosan to mannosan (LG/MN) and to galactosan (LG/GA) can help us to infer the $(OC/LG)_{bb}$ and $(EC/OC)_{bb}$ ratios from certain types of biomass fuel (Kawamura et al., 2012). Fig. 3 shows the source profile for LG/MN and

LG/GA ratios measured in emissions from controlled biomass burning experiments in previous studies (Cheng et al., 2013; Sun et al., 2019a; 2019b). There is a clear boundary of LG/MN ratios (~ 10) between softwood and hardwood burning (Kawamura et al., 2012). Using LG/GA ratios can help to identify the burning of straws, woods, needles and grasses, where most of the LG/GA ratios from burning of hardwood, straws and grasses are higher than 10, apparently different from those of needle plants. LG/MA and LG/GA ratios cannot be used to separate the burning of straws from hardwood, but the burning

of hardwood can be neglected considering the local forest types around Beijing (Cheng et al., 2013).

The measured LG/MN and LG/GA ratios in this study (Fig. 3) implied that the use of biofuels was mainly a mixture of softwood and crop straws. Compared with ratios in summer, LG/MN and LG/GA ratios in winter have a narrower range at both IAP and PG sites and most of them converge at values lower than 10, suggesting wood burning may be dominant in winter and the contribution of straw burning increases in summer. It has been reported that firewood burning emissions in





Beijing represent 47–90 % of the total biomass burning, and could contribute more than 80 % in winter (Zhou et al., 2017). Although the LG/MN and LG/GA ratios of maize burning and wheat burning are similar and cannot be used to distinguish the two, research into the yield of main crops and farming practices showed that the main straw burning in Beijing and nearby provinces may be associated with maize straw in October to November and wheat straw in May to July (Zhang et al., 2019).

These results suggest that softwood burning and straw burning are the main sources of aerosols from biomass burning in Beijing. Thus, EC from softwood burning and straw burning can be calculated as follows:

$EC_{nf} \approx EC_{bb} = EC_{wood} + EC_{straw}$

$= LG \times f_{wood} \times (OC/LG)_{wood} \times (EC/OC)_{wood} + LG \times f_{straw} \times (OC/LG)_{straw} \times (EC/OC)_{straw}$    (7)

where, $f_{wood}$ represents the fraction of LG from softwood burning, and $f_{straw}$ represents the fraction of LG from straw burning.

$f_{straw} = 1 - f_{wood}$, neglecting other sources of biomass burning. $f_{wood}$ can be expressed as,

$$f_{wood} = \frac{EC_{nf} - LG \times (OC/LG)_{straw} \times (EC/OC)_{straw}}{((OC/LG)_{wood} \times (EC/OC)_{wood} - (OC/LG)_{straw} \times (EC/OC)_{straw}) \times LG}$$    (8)

As $f_{wood}$ should in the range of 0~1, it can be used as the limits of selected EC/OC and OC/LG ratios from softwood and straw. The details of the ratio selection can be found in the Supplement. Once values of $f_{wood}$ are confirmed, OC from softwood ($OC_{wood}$) and straw burning ($OC_{straw}$) can be obtained by:

$OC_{wood} = LG \times f_{wood} \times (OC/LG)_{wood}$    (9)

$OC_{straw} = LG \times (1 - f_{wood}) \times (OC/LG)_{straw}$    (10)

$OC_{bb} = OC_{wood} + OC_{straw}$    (11)

The mass concentrations of $OC_{bb}$ and the contributions of $OC_{wood}$ and $OC_{straw}$ are shown in Fig. 4. The average concentration of $OC_{bb}$ in IAP winter was 2.7 ± 1.3 μg m$^{-3}$, with a contribution of 10.6 ± 1.7 % to total OC, which is about half that in PG

winter (4.8 ± 2.4 μg m$^{-3}$, 10.4 ± 1.5 %). The $OC_{bb}$ concentrations fell in summer (0.6 ± 0.7 μg m$^{-3}$ at IAP, 2.0 ± 0.8 μg m$^{-3}$ at PG), the contributions are 6.5 ± 5.2 % and 17.9 ± 3.5 %, respectively. The rural site always has a higher $OC_{bb}$, which is consistent with our previous discussions. $OC_{wood}$ normally dominated $OC_{bb}$ while the increase of $OC_{straw}$ fraction may be attributed to the local open burning activities. The contributions of $OC_{bb}$ in this study are slightly different from previous studies (Table 4), with lower percentage contributions in the winter sampling period. However, if applying a value of 12.2, a

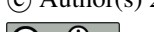



typical OC/LG ratio widely used in previous studies (Andreae and Merlet, 2001; Fu et al., 2012; Zhang et al., 2008a; Zhang et al., 2007), contributions of $OC_{bb}$ to OC would be 17.7 % and 26.3 % for IAP and PG in winter, 6.1 % and 10.8 % in summer, respectively. Compared with the results in this study, the ratio 12.2 may significantly overestimate the contributions of biomass burning in winter and underestimate the contributions in summer.

In addition, coal combustion may also emit LG, accounting for about 7 % of LG in Beijing (Yan et al., 2018; Zhang et al.,

2008b). Considering that the reported LG/MN and LG/GA ratios of coal combustion (both are in the range of 5–10) are close to those in our measurements, there may be a contribution from coal burning to levoglucosan, but if this amounts to 7 % of levoglucosan, the percentage contribution of biomass burning to OC in winter falls to 10.3±1.6 % and 10.2 ±1.5 % at IAP and PG respectively.

Besides the local emissions, regional transport of air masses can elevate the local OC and EC concentrations. To study the

influence of regional transport on biomass burning particles, back trajectories and fire spots are plotted in Fig. S1 and Fig. S2. As shown in Fig. S1, air masses at IAP and PG in winter originated in from Hebei, Shanxi and Inner Mongolia, and the fire spots showed low intensity of open burning activities along the air mass transport path during the measurement period. It suggests a less important effect of open burning through regional transport. During the summer, the open burning activities were greatly increased due to the harvesting of wheat (Zhang et al., 2019), which is confirmed by the fire spots distribution

in Fig. S2. Air masses from Hebei, Liaoning and Shandong provinces, which contain particles from wheat straw burning, may have enhanced the concentrations of OC and EC in Beijing. Considering the open burning of straw is linked with a sudden increase of LG in the ambient atmosphere, the high LG concentrations accompanied with low fire spot intensity suggests a strong local emission, while days with high fire spot intensities may also be affected by regional transport. Combining the analysis of fire spots intensity and the $OC_{wood}$ fractions can help to identify the influence of local emissions

and regional transport. The details of regional transport and sources of biomass burning are summarized in Table S2.

### 3.2.2 Other sources of OC and consistency with CMB and ASCM-PMF model results

More detailed source apportionment of OC can be achieved by combining [14]C analysis with primary OC/EC ratios for each source. $POC_f$ can be determined from $EC_f$ with primary fossil-fuel OC/EC emission ratios $(POC/EC)_f$. However, the $(POC/EC)_f$ ratios in previous studies (1.12–2.08 in winter, 0.40–0.77 in summer, Zhang et al., 2017) give much lower $POC_f$

values compared to CMB results (Fig. S5) even though mostly good correlations were found. In reality, $(POC/EC)_f$ ratios vary greatly according to combustion conditions, fuel types and even measurement method for OC and EC (Chow et al., 2001; Han et al., 2016), and it is very hard to accurately predict the $(POC/EC)_f$ ratio for a given area. Hence, we used the lowest $(OC/EC)_f$ ratios $(OC/EC)_{f, min}$ as the $(POC/EC)_f$ to estimate $POC_f$. Due to the limited number of samples for [14]C analysis, the measured lowest $(OC/EC)_f$ ratios may be higher than the ratios for the whole sampling period, which will result



in an overestimation of $POC_f$. It is necessary to evaluate $(OC/EC)_{f, min}$ ratios for the whole sampling period. The evaluation method is described in the Supporting Information. In the same way, primary OC from non-fossil sources ($POC_{nf}$) can be calculated from $EC_{nf}$ and lowest $(OC/EC)_{nf}$ ratios, thereby concentrations of secondary OC from fossil sources ($SOC_f$), non-fossil sources ($SOC_{nf}$) and OC from cooking ($OC_{ck}$) can be obtained by equations in Table 1. The averaged source apportionment results are presented in Table 5.

Primary fossil-derived OC is mainly from coal combustion and traffic emissions in China. However, it cannot be distinguished by $^{14}C$ analysis. OC/EC ratios from coal combustion and traffic emissions are dependent on various factors, such as the types of coal, stoves, engines, the vehicle operating modes and test method. Typical OC/EC ratios of coal combustion and traffic emissions in Beijing are $2.38 \pm 0.44$ and $0.85 \pm 0.16$, respectively (Ni et al., 2018). An upper limit of POC from traffic emissions ($POC_{tr}$) can be obtained by multiplying $EC_f$ with the $(OC/EC)_{tr}$ ratio ($0.85 \pm 0.16$) considering

all $EC_f$ to come from traffic emissions. A lower limit of POC from coal combustion ($POC_{cc}$) is obtained by subtracting $POC_{tr}$ from $POC_f$. Such calculation shows that $POC_{cc}$ dominated POC at both sites in winter and summer campaigns. The maximum contribution of $POC_{tr}$ to OC was 7.3 % and 5.7 % in winter, and 6.8 % and 8.9 % in summer, for IAP and PG respectively, and $POC_{cc}$ contributed at least 28.5 % and 28.4 % to OC for IAP and PG in winter, and 22.2 % and 20.0 % in summer.

This is a relatively crude method for source apportionment of primary OC from fossil and non-fossil sources. Therefore, further comparisons with results from application of a CMB method and from application of PMF to ACSM data were conducted to understand the uncertainties in source apportionment from different methods. The source contributions to OC at the IAP and PG sites in winter and summer from the CMB model (Xu et al., 2020b; Wu et al., 2020) are presented in Table 5. In brief, seven primary OC sources were apportioned, including emissions from vegetative detritus, biomass

burning, cooking, gasoline vehicles, diesel engines, industrial coal combustion, residential coal combustion, along with Other (secondary) OC. Among these sources, coal combustion (the total of residential and industrial coal combustion) accounted for 32.6 % to OC at IAP in winter and 40.0 % to OC at PG in summer, while other OC dominated OC at IAP in summer and at PG in winter, with contributions of 48.2 % and 32.5 %, respectively.

For comparison, OC from gasoline vehicles, diesel engines, industrial coal combustion and residential coal combustion

resolved by the CMB model are summed up as $POC_f$, and OC from vegetative detritus, biomass burning and cooking are summed up as $POC_{nf}$. Correlations of different OC sources from the extended Gelencsér method (EG method) and from the CMB model are shown in Fig. 5. Good correlations were found for $POC_f$, $POC_{nf}$, SOC and $OC_{bb}$ despite the combination of sites and seasons ($R^2$=0.96, 0.74, 0.85 and 0.91). The EG method reported lower $POC_f$, $POC_{nf}$, and $OC_{bb}$ values than those from CMB with slopes of 0.77, 0.66 and 0.53, respectively. More specifically, $OC_{bb}$ by the EG method is 51 % of that by





CMB in winter, but 1.33 times higher than CMB in summer (Fig. S6). The main discrepancy within the apportionment of $OC_{bb}$ is caused by different parameters for the calculations. As the CMB model used source profiles from three major types of cereal straw (wheat, corn, and rice) and two types of wood (pine and mixed wood), it may lead to an overestimation of OC from straw burning. Closer values of $POC_{nf}$ and SOC were found between the two methods in summer, when samples are almost all belonging to the non-haze period (Fig. S6, Fig. S7). It indicated that the EG method may perform better when OC concentrations are low. Although poor agreement of $OC_{ck}$ was found between the EG method and the CMB model, the former correlated better with results from the application of PMF to AMS/ACSM data (slope=0.74, $R^2$=0.61). It had previously been shown that discrepancies existed between CMB and PMF model in the quantification of $OC_{ck}$. ACSM-PMF may overestimate $OC_{ck}$ by approximately 2 times (Reyes-Villegas, et al., 2018; Yin et al., 2015), whereas, CMB may not be sensitive enough to the source profile of cooking aerosols (Abdullahi et al., 2018). Overall, the EG method resolves primary and secondary sources of OC well.

Fractions of $POC_f$, $SOC_f$, $OC_{bb}$, $OC_{ck}$ and $SOC_{nf}$ in OC are shown in Fig. 6. $POC_f$ was the largest contributor to OC at both sites through winter and summer. Comparable contributions of $POC_f$ were observed at the urban and rural sites, which reached to $35.8 \pm 10.5$ % and $34.1 \pm 8.7$ % in wintertime, and fell to $28.9 \pm 7.4$ % and $29.1 \pm 9.4$ % in summer, respectively. Pronounced $POC_f$ in wintertime especially during haze period implied a significant elevation of coal combustion and traffic emissions. Fossil and non-fossil sources of SOC are distinguished by the EG method in this study for the first time. Average contributions to OC from $SOC_f$ are higher in winter. They decreased from $32.0 \pm 12.5$ % to $25.2 \pm 7.6$ % at IAP, and from $25.2 \pm 10.4$ % to $21.0 \pm 14.4$ % at PG from winter to summer. The contributions of $SOC_{nf}$ are slightly greater in summer ($18.0 \pm 2.9$ %, $22.0 \pm 17.6$ % for IAP in winter and summer and $16.9 \pm 10.8$ %, $21.9 \pm 16.4$ % for PG in winter and summer, respectively). Significant contributions of $SOC_f$ in the winter sampling period indicated a greater fraction of $OC_f$ from ageing and oxidation. The elevated contributions of $SOC_{nf}$ (as a percentage) in the summer sampling period may be assigned to the reduced coal combustion and enhanced biogenic-derived SOC formation. Similar variations of $SOC_f$ and $OC_{onf}$ (all OC from non-fossil sources excluding $OC_{bb}$) between winter and summer were found in the urban area of Beijing by Zhang et al. (2016). The total SOC accounts for $50.0 \pm 12.3$ % and $42.0 \pm 11.7$ % of OC for IAP and PG site in winter, demonstrating an important role of secondary formation processes especially at the urban site. The average contributions of $OC_{ck}$ were $3.6 \pm 2.7$ % and $13.4 \pm 5.8$ % in winter for IAP and PG, and $17.4 \pm 12.5$ % and $10.4 \pm 6.7$ % in summer, close to those estimated in previous studies ($19 \pm 4$ %, Zhang et al., 2017). The slightly lower value of the $OC_{ck}$ contribution in winter at the IAP site (Fig. 5) was due to $OC_{bb}$ being the overwhelming contributor to $POC_{nf}$.

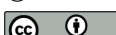



### 3.2.3 Correlations of WINSOC, WSOC with POC and SOC

In order to better understand the origins and formation mechanism of OC, the correlations among WINSOC, WSOC, POC,

SOC, $OC_{bb}$ and $OC_{ck}$ were investigated (Fig. 7). The regression slopes and correlation coefficients among them are summarized in Table S5. $WINSOC_f$ is usually seen as a proxy of primary fossil-derived OC in many previous studies (Liu et al., 2016; Miyazaki et al., 2006), and in our study $WINSOC_f$ correlated well with calculated $POC_f$ by the EG method ($R^2$ of 0.97, 0.93, 0.97 and 0.82 for IAP and PG in winter and summer, respectively); good correlations were also observed between $WINSOC_f$ and $SOC_f$ with slopes of 1.54, 1.27, 0.99 and 0.99 for IAP and PG in winter and summer (corresponding $R^2$ is

0.96, 0.89, 0.91 and 0.43). The high $WINSOC_f$ to $SOC_f$ ratios implied a non-negligible fraction of $WINSOC_f$ in $SOC_f$. Moreover, the ratios of $WINSOC_f$ to $POC_f$ decreased from winter to summer compared to the $WSOC_f$ to $POC_f$ ratios increasing, indicating a non-negligible fraction of $WSOC_f$ in $POC_f$ in summer. $WSOC_{nf}$ and $WINSOC_{nf}$ show good correlations with $SOC_{nf}$ with larger $WINSOC_{nf}$ to $SOC_{nf}$ ratios in winter. The lower water solubility of $SOC_f$ and $SOC_{nf}$ in winter may be due to them originating from the less oxidized semi-volatile POC from wood burning and anthropogenic

emission at low temperatures (Favez et al., 2008; Sciare et al., 2011). Weak photochemical activity would also lead to the formation of less oxidized SOC which is more water-insoluble (Donahue et al., 2006; Robinson et al., 2007). Significant contributions of WINSOC to SOC have been reported in France (Sciare et al., 2011) and Switzerland (Zhang et al., 2016). $OC_{bb}$ correlated well with $WSOC_{nf}$ ($R^2$ of 0.94, 0.92, 0.93 and 0.93, at IAP and PG in winter and summer) as it is mainly composed of polar and highly oxygenated compounds (Miyazaki et al., 2006). However, more pronounced $WINSOC_{nf}$ in

$OC_{bb}$ was found in winter especially at the rural site. It seems that more water-insoluble fractions were observed in primary OC ($POC_f$, $POC_{nf}$, $OC_{bb}$ and $OC_{ck}$) at the rural site in both winter and summer. It implies the emitted primary OC at the rural site was probably fresher, and hence less aged and oxidized. On the other hand, the source emission profile at rural sites may be different from urban sites, with more heavy-duty diesel trucks with a high content of water-insoluble OC emitted in rural areas.

### 4 CONCLUSIONS

Measurements of $PM_{2.5}$, OC, EC and biomass burning tracers were conducted at both urban and rural sites of Beijing from 10 November to 11 December 2016 and from 22 May to 24 June 2017, accompanied by the [14]C analysis of 25 selected samples. On most days, fossil sources dominated EC at IAP and PG in winter and summer, with contributions of 45.9–71.7 % at IAP and 48.2–76.6 % at PG. The fossil sources of OC contribute 34.7–75.0 % and 39.3–66.9 % for IAP and PG with

non-fossil fractions of OC elevated in summer. An extended Gelencsér method using the [14]C measurements was applied for the first time to estimate fossil and non-fossil sources of primary and secondary OC, as well as OC from biomass burning and cooking ($POC_f$, $SOC_f$, $POC_{nf}$, $SOC_{nf}$, $OC_{bb}$ and $OC_{ck}$, respectively). Fossil-derived POC is the major contributor during winter and summer at both sites. Fossil-derived SOC contributed more in winter especially at the urban site, with average

contributions (to OC) of $32.0 \pm 12.5$ % and $25.2 \pm 7.6$ % for IAP, $25.2 \pm 10.4$ % and $21.1 \pm 14.4$ % for PG in winter and summer, respectively. The contribution of $SOC_{nf}$ increased in summer, which is probably associated with formation from biogenic emissions. A study of relationships among levoglucosan, mannosan and galactosan showed that biomass burning was mainly from softwood combustion and straw burning. The extended Gelencsér method using $^{14}C$ data provided a more robust calculation of $OC_{bb}$. The contributions of $OC_{bb}$ to OC were $10.6 \pm 1.7$ % and $10.4 \pm 1.5$ % for IAP and PG in winter, $6.5 \pm 5.2$ % and $17.9 \pm 3.5$ % for IAP and PG in summer. Correlations among WINSOC, WSOC and POC, SOC showed that $WINSOC_f$ and $WINSOC_{nf}$ were the main components of $POC_f$ and $POC_{nf}$, respectively. However, large fractions of WINSOC were found in both $SOC_f$ and $SOC_{nf}$ especially at the rural site, and the contributions of water-insoluble OC decreased from winter to summer, with more WSOC formed under favourable conditions in summer. Although derived from a limited number of samples, our study reflected the different formation mechanisms of SOC between winter and summer, and between the urban and rural area. It also confirms the feasibility of a new approach of direct source apportionment of carbonaceous aerosol, which was found to compare generally well with the commonly used Chemical Mass Balance and AMS/ACSM-PMF methods.

**Data accessibility**

Data supporting this publication are openly available from the UBIRA eData repository at https://doi.org/10.25500/edata.bham.00000572.

**Author contribution**

Z.S. and R.M.H. conceived the research. T.V.V. and D.L. conducted the aerosol sampling and laboratory-based OC/EC analyses. D.L. and L.J.L carried out the GC-MS analysis. P.Q.F. supervised GC-MS laboratory work. A.V., V.M. and G.S. carried out the $^{14}C$ analysis. S.S. and A.S.H.P supervised the $^{14}C$ analysis. X.W. and J.X. conducted the CMB modelling at PG and IAP sites, respectively. S.H. conducted the analysis of the extended Gelencsér (EG) method incorporating $^{14}C$ data. Y.S. provided the AMS-PMF data. S.H. and D.L. drafted the paper.

**Competing interests**

The authors have no conflict of interests.

**Special issue statement**

This article is part of the special issue "In-depth study of air pollution sources and processes within Beijing and its surrounding region (APHH-Beijing) (ACP/AMT inter-journal SI)". It is not associated with a conference.



**Acknowledgements**

This research was funded by the Natural Environment Research Council (Grant No: NE/N007190/1, NE/R005281/1). We thank Bill Bloss, Leigh Crilley, Louisa Kramer from the University of Birmingham, Siyao Yue, Liangfang Wei, Hong Ren, Qiaorong Xie, Wanyu Zhao, Linjie Li, Ping Li, Shengjie Hou, Qingqing Wang, Pingqing Fu and Yele Sun from Institute of Atmospheric Physics, Rachel Dunmore, Ally Lewis, Jacqui Hamilton and James Lee from the University of York, Kebin He and Xiaoting Cheng from Tsinghua University, and James Allan and Hugh Coe from the University of Manchester, Yiqun Han, Hanbing Zhang from King's College London, and Tong Zhu from Peking University for providing logistic and scientific support for the field campaigns. PQF acknowledges funding from the Royal Society – NSFC Advanced Newton Fellowship (NAF\R1\191220).

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





**TABLE LEGENDS:**

**Table 1**      Equations for [14]C-based source apportionment, see Sect. 3.2 for the details.

**Table 2**      Statistical summary of concentrations, ratios and meteorological parameters at the IAP and PG sites during winter and summer campaigns.

**Table 3**      Pearson correlations of species at IAP and PG sites.

**Table 4**      Comparison of LG concentrations and OC from biomass burning in this study and related literature.

**Table 5**      Source contribution estimates ($\mu g\ m^{-3}$) for OC at IAP and PG in winter and summer.

**FIGURE LEGENDS:**

**Figure 1**      Time series of $PM_{2.5}$ and its major components at IAP and PG during winter (right) and summer (left).

**Figure 2**      Time series of concentrations of $WSOC_f$, $WSOC_{nf}$, $WINSOC_f$, $WINSOC_{nf}$, $EC_f$ and $EC_{nf}$ (left) and their relative contributions to TC (right) at IAP and PG.

**Figure 3**      Scatter plot of LG/MN vs. LG/GA from different types of biomass burning emissions (Cheng et al., 2013; Sun et al., 2019b; Sun et al., 2019a), including those measured in $PM_{2.5}$ samples at IAP and PG during winter and summer. The range of LG/MA is 7.66–13.41 and 5.48–22.40 for IAP in winter and summer, 2.45–60.20 and 3.27–17.71 for PG in winter and summer, while the range of LG/GA is 3.98–6.92, 2.78–101.43, 1.97–34.66 and 1.51–13.39, respectively.

**Figure 4**      Concentrations of OC from softwood ($OC_{wood}$) vs. OC from straw ($OC_{straw}$) at IAP (upper) and PG (lower) and variations of $OC_{wood}$ fractions.

**Figure 5**      Correlations of OC sources from extended Gelencsér method with those from CMB model. EG denotes extended Gelencsér method, (a): primary OC from fossil sources, (b): primary OC from non-fossil sources, (c): secondary OC, (d): OC from biomass burning, (e): OC from cooking, (f): correlations of $OC_{ck}$ from extended Gelencsér method and AMS/ACMS-PMF model (AMS for IAP and ACMS for PG).

**Figure 6**      Fractions of each source (i.e., $POC_f$, $SOC_f$, $OC_{bb}$, $OC_{ck}$ and $SOC_{nf}$) in OC based on the extended Gelencsér method. f: fossil fuel sources, nf: non-fossil sources, bb: biomass burning, ck: cooking. The box denotes





the 25th (lower line), 50th (middle line), and 75th (top line) percentiles; the solid squares within the box denote the mean values; the end of the vertical bars represents the 10th (below the box) and 90th (above the box) percentiles; and the solid dots denote maximum and minimum values.

**Figure 7**      Correlations of WINSOC, WSOC with POC, SOC at IAP and PG sites in winter and summer. The slopes and correlation coefficients are summarized in Table S5.



**Table 1.** Equations for $^{14}$C-based source apportionment, see Sect. 3.2 for the details.

|  | Extended Gelencsér method |
| --- | --- |
| $EC_{bb}$ | $\approx EC_{nf}$, measured by $^{14}$C |
| $EC_f$ | measured by $^{14}$C |
| $OC_{nf}$ | measured by $^{14}$C |
| $OC_f$ | measured by $^{14}$C |
| $POC_f$ | $EC_f \times (OC/EC)_{f,\,min}$ |
| $SOC_f$ | $OC_f - POC_f$ |
| $POC_{nf}$ | $EC_{nf} \times (OC/EC)_{nf,\,min}$ |
| $SOC_{nf}$ | $OC_{nf} - POC_{nf}$ |
| $OC_{bb}$ | $EC_{nf} \times (OC/EC)_{bb} = LG \times (OC/LG)_{bb}$ |
| $OC_{bio}$ | Ignored |
| $OC_{ck}$ | $POC_{nf} - OC_{bb}$ |




**Table 2.** Statistical summary of concentrations, ratios and meteorological parameters at the IAP and PG sites during winter and summer campaigns.

| Compound/ Meteorological parameters | IAP | | | | | | | | PG | | | | | | | |
| --- | --- | --- | --- | --- | --- | --- | --- | --- | --- | --- | --- | --- | --- | --- | --- | --- |
| | Winter (n = 32) | | | | Summer (n = 34) | | | | Winter (n = 32) | | | | Summer (n = 34) | | | |
| | Mean | Std | Min | Max | Mean | Std | Min | Max | Mean | Std | Min | Max | Mean | Std | Min | Max |
| $PM_{2.5}$ (µg m$^{-3}$) | 97.7 | 75.3 | 8.1 | 328.7 | 30.2 | 14.8 | 12.2 | 78.8 | 99.7 | 77.8 | 13.3 | 294.3 | 27.5 | 12.9 | 11.6 | 70.3 |
| OC (µg m$^{-3}$) | 20.5 | 12.2 | 3.9 | 48.8 | 6.4 | 2.3 | 1.8 | 12.7 | 33.2 | 22.0 | 3.8 | 85.0 | 7.7 | 3.4 | 1.8 | 17.9 |
| EC (µg m$^{-3}$) | 3.3 | 2.0 | 0.3 | 6.6 | 0.9 | 0.4 | 0.2 | 1.7 | 3.7 | 2.3 | 0.3 | 9.8 | 1.2 | 0.7 | 0.1 | 3.1 |
| OC/EC | 6.9 | 2.4 | 4.1 | 14.9 | 7.6 | 2.2 | 4.6 | 14.8 | 9.0 | 1.9 | 6.2 | 14.6 | 9.0 | 6.7 | 4.4 | 28.3 |
| LG (ng m$^{-3}$) | 310.7 | 196.0 | 20.4 | 634.7 | 27.9 | 29.6 | 2.9 | 179.6 | 634.3 | 483.2 | 52.1 | 1796.1 | 74.0 | 34.2 | 29.9 | 219.8 |
| MN (ng m$^{-3}$) | 32.4 | 20.7 | 2.3 | 65.8 | 2.6 | 2.4 | 0.4 | 14.2 | 81.1 | 63.2 | 5.7 | 224.4 | 10.3 | 3.7 | 4.2 | 19.5 |
| GA (ng m$^{-3}$) | 60.3 | 39.7 | 4.6 | 138.6 | 4.3 | 2.5 | 0.1 | 12.0 | 190.2 | 172.9 | 7.8 | 723.1 | 14.2 | 7.6 | 4.1 | 38.0 |
| $K^+$ (µg m$^{-3}$) | 1.2 | 1.0 | 0.2 | 3.8 | 0.4 | 0.4 | 0.1 | 2.0 | 1.6 | 1.3 | 0.1 | 5.5 | 0.5 | 0.3 | 0.0 | 1.7 |
| WS (m s$^{-1}$) | 1.4 | 1.2 | 1.5 | 5.6 | 3.6 | 0.8 | 1.9 | 5.6 | 1.4 | 0.8 | 0.5 | 3.4 | 1.1 | 0.6 | 0.4 | 2.5 |
| RH (%) | 41.8 | 15.1 | 18.2 | 80.9 | 43.7 | 16.6 | 18.4 | 85.9 | 51.7 | 13.2 | 32.0 | 78.1 | 50.2 | 14.3 | 30.5 | 84.0 |
| T (ºC) | 7.0 | 3.3 | 0.4 | 14.7 | 24.6 | 3.6 | 16.7 | 31.7 | 1.0 | 3.2 | -4.2 | 8.4 | 23.3 | 3.7 | 15.6 | 31.7 |

Std, standard deviation; LG, MN and GA referred to levoglucosan, mannosan and galactosan, respectively; WS, wind speed;

RH, relative humidity, T, temperature.






**Table 3.** Pearson correlations of species at IAP and PG sites.

| IAP winter | PM$_{2.5}$ | OC | EC | K$^+$ | LG | MN | GA |
|---|---|---|---|---|---|---|---|
| PM$_{2.5}$ | 1.00 | | | | | | |
| OC | 0.91 | 1.00 | | | | | |
| EC | 0.86 | 0.92 | 1.00 | | | | |
| K$^+$ | 0.74 | 0.67 | 0.71 | 1.00 | | | |
| LG | 0.56 | 0.60 | 0.74 | 0.51 | 1.00 | | |
| MN | 0.52 | 0.57 | 0.72 | 0.48 | 0.99 | 1.00 | |
| GA | 0.52 | 0.55 | 0.70 | 0.52 | 0.98 | 0.97 | 1.00 |
| PG winter | PM$_{2.5}$ | OC | EC | K$^+$ | LG | MN | GA |
| PM$_{2.5}$ | 1.00 | | | | | | |
| OC | 0.95 | 1.00 | | | | | |
| EC | 0.85 | 0.93 | 1.00 | | | | |
| K$^+$ | 0.88 | 0.78 | 0.70 | 1.00 | | | |
| LG | 0.89 | 0.89 | 0.81 | 0.86 | 1.00 | | |
| MN | 0.85 | 0.85 | 0.82 | 0.84 | 0.94 | 1.00 | |
| GA | 0.88 | 0.85 | 0.74 | 0.84 | 0.95 | 0.94 | 1.00 |
| IAP summer | PM$_{2.5}$ | OC | EC | K$^+$ | LG | MN | GA |
| PM$_{2.5}$ | 1.00 | | | | | | |
| OC | 0.72 | 1.00 | | | | | |
| EC | 0.34 | 0.79 | 1.00 | | | | |
| K+ | 0.65 | 0.64 | 0.36 | 1.00 | | | |
| LG | 0.52 | 0.59 | 0.36 | 0.85 | 1.00 | | |
| MN | 0.47 | 0.55 | 0.37 | 0.80 | 0.97 | 1.00 | |
| GA | 0.41 | 0.59 | 0.54 | 0.59 | 0.79 | 0.85 | 1.00 |
| PG summer | PM$_{2.5}$ | OC | EC | K$^+$ | LG | MN | GA |
| PM$_{2.5}$ | 1.00 | | | | | | |
| OC | 0.60 | 1.00 | | | | | |
| EC | 0.41 | 0.75 | 1.00 | | | | |
| K+ | 0.65 | 0.80 | 0.57 | 1.00 | | | |
| LG | 0.42 | 0.46 | 0.32 | 0.51 | 1.00 | | |
| MN | 0.47 | 0.22 | -0.03 | 0.17 | 0.65 | 1.00 | |
| GA | 0.53 | 0.30 | 0.03 | 0.22 | 0.33 | 0.55 | 1.00 |

LG, MN and GA referred to levoglucosan, mannosan and galactosan, respectively.



**Table 4.** Comparison of LG concentrations and OC from biomass burning in this study and related literature.

| Site | Site Type | Sampling Time | LG ng m$^{-3}$ | OC$_{bb}$ contribution | OC/LG used for estimation | Reference |
|---|---|---|---|---|---|---|
| Beijing | Urban | Winter (16 Nov-12 Dec 2016) | $310.7 \pm 196.0$ | $10.6 \pm 1.7\%$ | | This Study |
| Beijing | Urban | Winter in 2013-2014 | 189, 36.1–491 | 12.2%, 3.61–19.5% | 12.29 | Kang et al., 2018 |
| Beijing | Urban | Winter in 2012-2013 | 361, 171–730 | 16.6%, 6.06–35.2% | 12.20–12.50 | Li et al., 2018 |
| Wuhan | Urban | Winter (9 January-6 February 2013) | $950 \pm 421$ | $21\% \pm 9\%$ | $7.76 \pm 1.47$ | Liu et al., 2016b |
| Beijing | Urban | Summer (22 May-22 Jun 2017) | $27.9 \pm 29.6$ | $6.5 \pm 5.2\%$ | | This Study |
| Beijing | Urban | Summer (9 June-8 July 2014) | $56.37 \pm 55.48$ | $14.8 \pm 9.4\%$ | 20.83 | Yan et al., 2019 |
| Beijing | Urban | Summer in 2014 | 12.4, 0.84–26.8 | 2.73%, 0.28–5.60% | 12.29 | Kang et al., 2018 |
| Beijing | Urban | Summer in 2012 | 61.8, 13.9–317 | 8.39%, 2.64–12.5% | 12.20–12.50 | Li et al., 2018 |
| Beijing | Rural | Winter (16 Nov-12 Dec 2016) | $634.3 \pm 483.2$ | $10.4 \pm 1.5\%$ | | This Study |
| Xi'an | Rural | Winter (17-26 January 2014) | $930 \pm 320$ in PM$_{0.133}$ | 24%, 19–32% | 12.2 | Zhu et al., 2017 |
| Beijing | Rural | Summer (22 May-22 Jun 2017) | $74.0 \pm 34.2$ | $17.9 \pm 3.5\%$ | | This Study |
| Zhengzhou | Suburban | Summer, BB episode in 2-21 June 2015 | 460–1230 | 47.20% | 13.51 | Chen et al., 2018 |
| Zhengzhou | Suburban | Summer, Non-BB in 2-21 June 2015 | 200–290 | 13.90% | 11.9 | Chen et al., 2018 |
| Hebei | Rural | Summer (9 June-8 July 2014) | $205.94 \pm 304.35$ | $31.3 \pm 18.8\%$ | 20.83 | Yan et al., 2019 |






**Table 5.** Source contribution estimates (µg m⁻³) for OC at IAP and PG in winter and summer.

| Sources | Winter | | | Winter | | | Summer | |
|---|---|---|---|---|---|---|---|---|
| **EG method results** | IAP haze (n=5) | IAP non-haze (n=2) | Winter (n=7) | PG haze (n=5) | PG non-haze (n=2) | Winter (n=7) | IAP (n=6) | PG (n=5) |
| Fossil-derived POC (POC$_f$) | 13.5±3.9 | 3.0±3.5 | 10.5±6.2 | 20.4±2.5 | 5.0±1.3 | 16.0±7.8 | 2.3±0.8 | 3.3±1.9 |
| Biomass burning (OC$_{bb}$) | 3.4±0.9 | 1.2±0.9 | 2.7±1.3 | 6.1±1.7 | 1.9±0.5 | 4.8±2.4 | 0.6±0.7 | 2.0±0.8 |
| Cooking (OC$_{ck}$) | 1.3±0.5 | 0.4±0.7 | 1.1±0.7 | 6.8±3.7 | 3.0±0.7 | 5.8±3.6 | 1.1±0.4 | 0.9±0.4 |
| Fossil-derived SOC (SOC$_f$) | 9.5±3.5 | 3.3±1.3 | 7.7±4.2 | 16.8±10.4 | 4.6±2.6 | 13.3±10.4 | 2.0±0.9 | 2.1±1.5 |
| Non-fossil-derived SOC (SOC$_{nf}$) | 6.2±2.1 | 1.5±1.0 | 4.8±2.9 | 11.9±9.2 | 1.9±1.0 | 9.1±9.0 | 2.2±2.1 | 3.1±3.1 |
| **CMB results** | | | | | | | | |
| Gasoline vehicle | 2.8±1.2 | 1.4±0.2 | 2.4±1.2 | 1.6±1.1 | 0.5±0.2 | 1.3±1.0 | 0.4±0.1 | 0.1±0.0 |
| Diesel vehicle | 1.3±2.1 | 0.0±0.0 | 0.9±1.8 | 11.4±3.9 | 1.9±1.5 | 8.7±5.7 | 0.1±0.2 | 0.6±0.3 |
| Industrial CC | 5.5±3.9 | 0.6±0.3 | 4.1±4.0 | 4.9±2.4 | 1.6±0.2 | 4.0±2.5 | 2.1±0.4 | 4.2±2.6 |
| Residential CC | 5.7±4.8 | 2.1±2.3 | 4.6±4.4 | 8.6±4.7 | 3.3±1.4 | 7.1±4.7 | 0.2±0.1 | 0.4±0.2 |
| Vegetative Detritus | 0.1±0.1 | 0.1±0.0 | 0.1±0.1 | 3.3±4.7 | 0.5±0.3 | 2.5±4.1 | 0.2±0.1 | 0.3±0.3 |
| Biomass Burning | 5.5±1.9 | 1.8±1.4 | 4.4±2.5 | 11.6±4.3 | 3.0±0.8 | 9.2±5.5 | 0.6±0.8 | 1.2±0.7 |
| Cooking | 3.6±3.3 | 0.7±0.5 | 2.8±3.1 | 0.2±0.2 | 0.6±0.1 | 0.3±0.3 | 0.6±0.3 | 0.6±0.4 |
| Other OC | 9.4±3.6 | 2.7±2.6 | 7.4±5.7 | 20.2±7.4 | 5.0±2.3 | 15.9±9.6 | 4.0±2.2 | 4.1±2.5 |
| Total OC | 33.8±8.6 | 9.4±7.4 | 26.8±14.2 | 62.0±19.4 | 16.4±6.1 | 48.9±27.3 | 8.3±3.2 | 11.5±4.9 |

CMB results are from Wu et al., 2020 and Xu et al., 2020

CC: coal combustion; Other OC is calculated by subtracting seven primary sources of OC from total measured OC.






**Figure 1.** Time series of PM$_{2.5}$ and its major components at IAP and PG during winter (right) and summer (left).





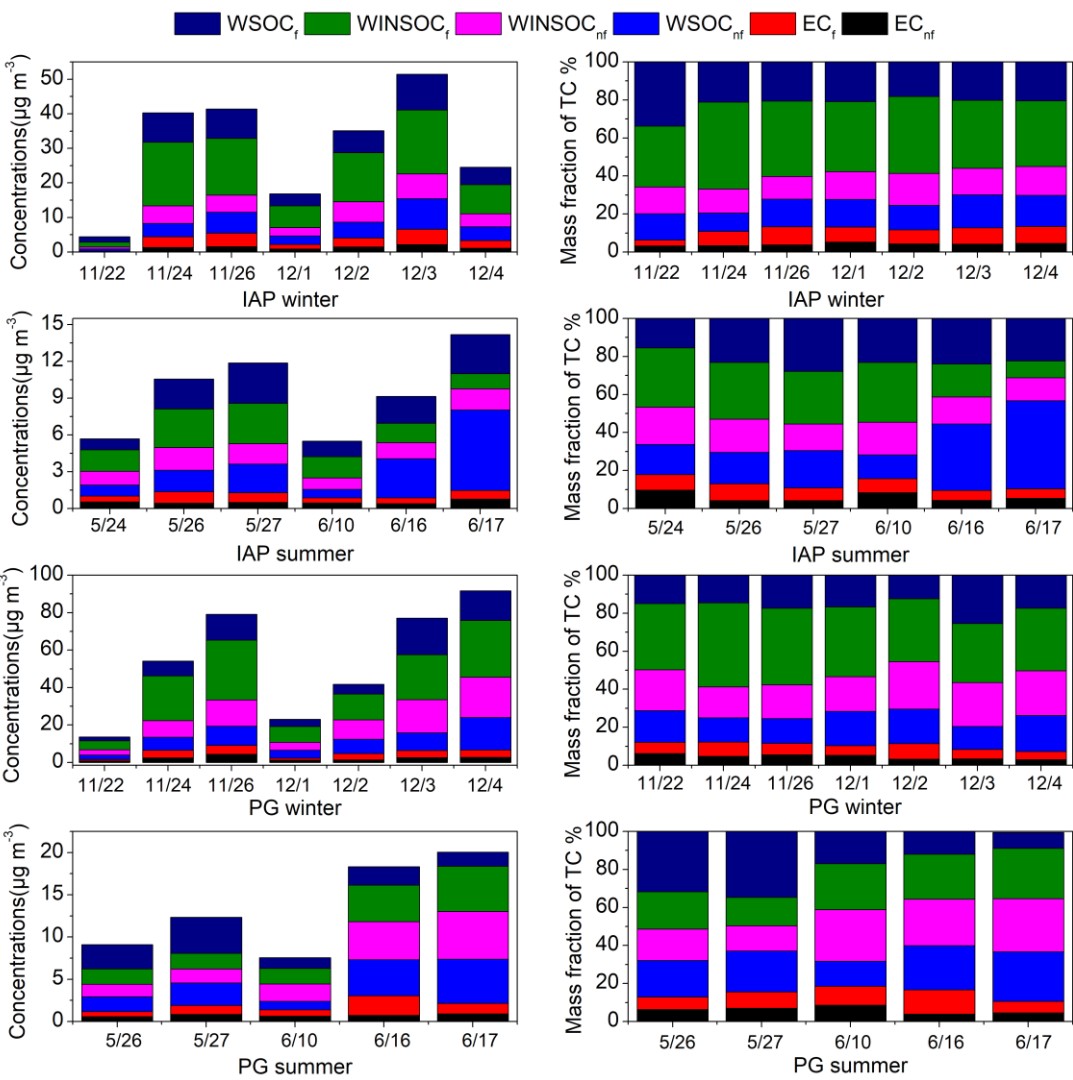

**Figure 2.** Time series of concentrations of $WSOC_f$, $WSOC_{nf}$, $WINSOC_f$, $WINSOC_{nf}$, $EC_f$ and $EC_{nf}$ (left) and their relative contributions to TC (right) at IAP and PG.





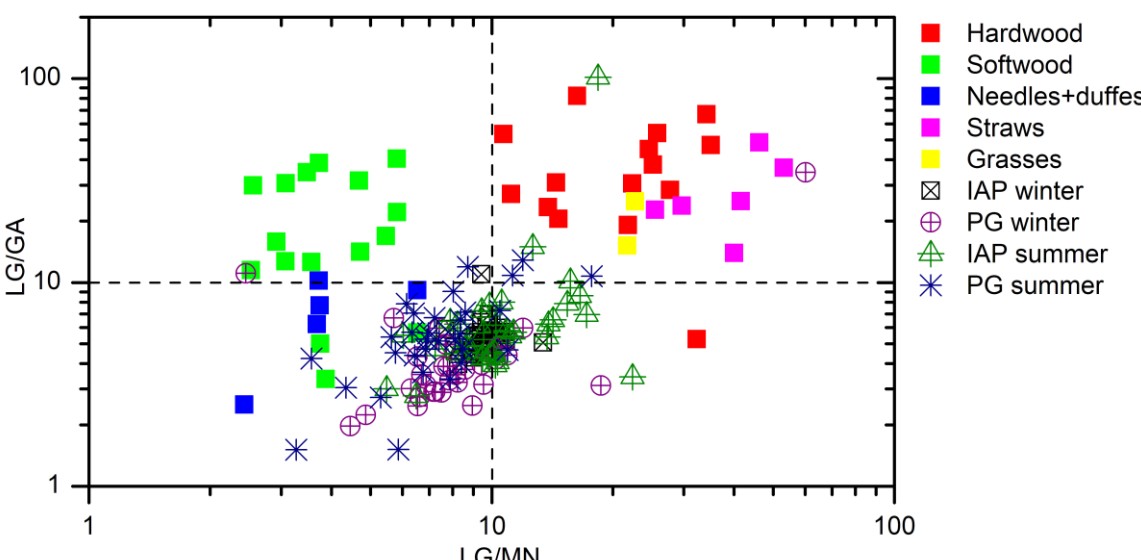

**Figure 3.** Scatter plot of LG/MN vs. LG/GA from different types of biomass burning emissions (Cheng et al., 2013; Sun et al., 2019b; Sun et al., 2019a), including those measured in $PM_{2.5}$ samples at IAP and PG during winter and summer. The range of LG/MA is 7.66–13.41 and 5.48–22.40 for IAP in winter and summer, 2.45–60.20 and 3.27–17.71 for PG in winter and summer, while the range of LG/GA is 3.98–6.92, 2.78–101.43, 1.97–34.66 and 1.51–13.39, respectively.





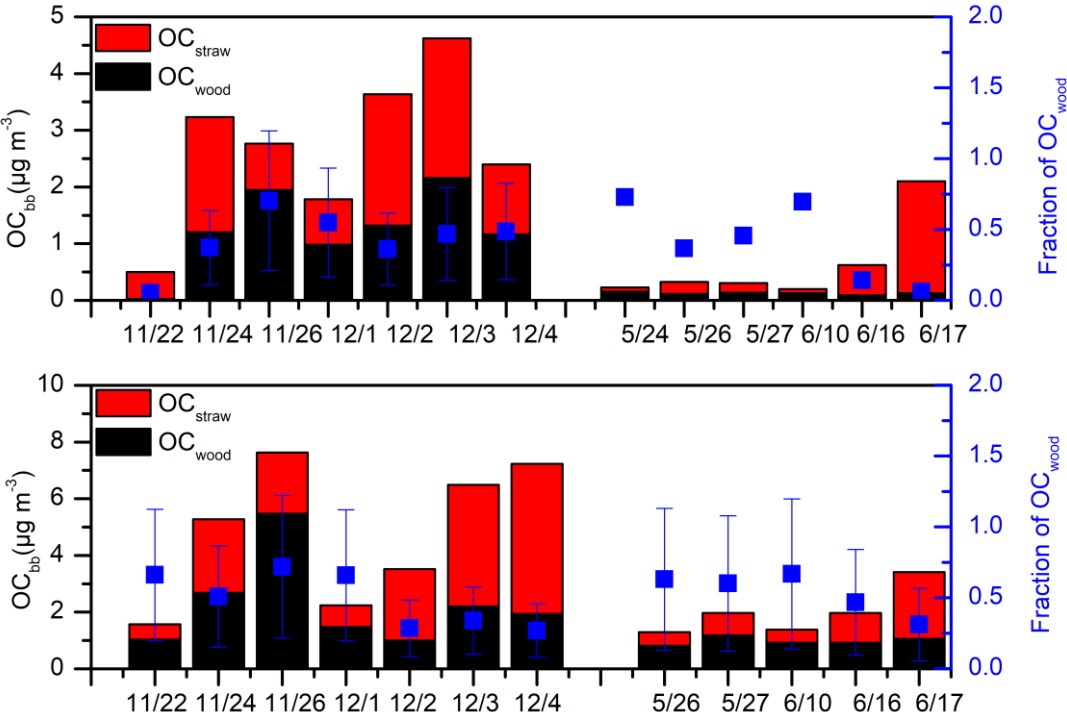

**Figure 4.** Concentrations of OC from softwood ($OC_{wood}$) vs. OC from straw ($OC_{straw}$) at IAP (upper) and PG (lower) and

variations of $OC_{wood}$ fractions.





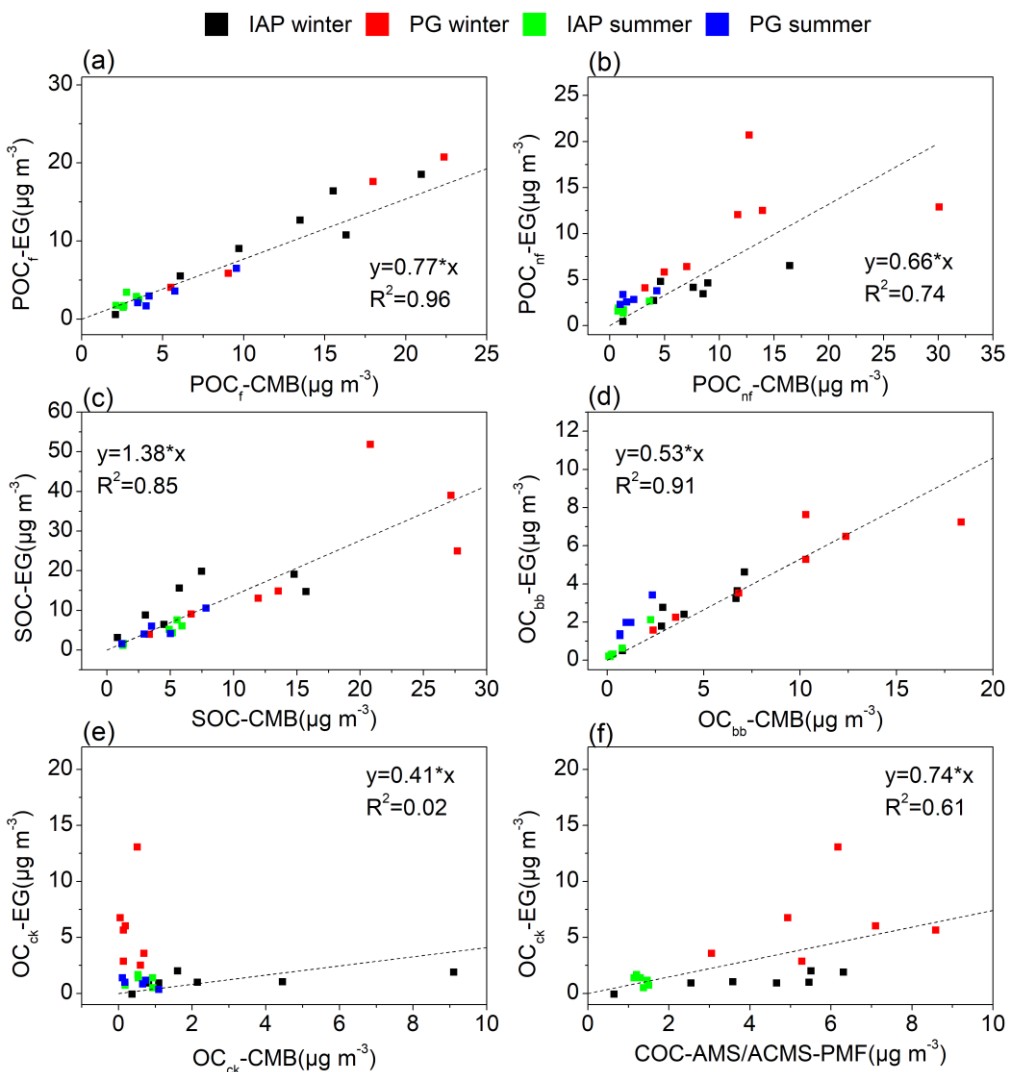

**Figure 5.** Correlations of OC sources from extended Gelencsér method with those from CMB model. EG denotes extended Gelencsér method, (a): primary OC from fossil sources, (b): primary OC from non-fossil sources, (c): secondary OC, (d): OC from biomass burning, (e): OC from cooking, (f): correlations of $OC_{ck}$ from extended Gelencsér method and AMS/ACMS-PMF model (AMS for IAP and ACMS for PG).





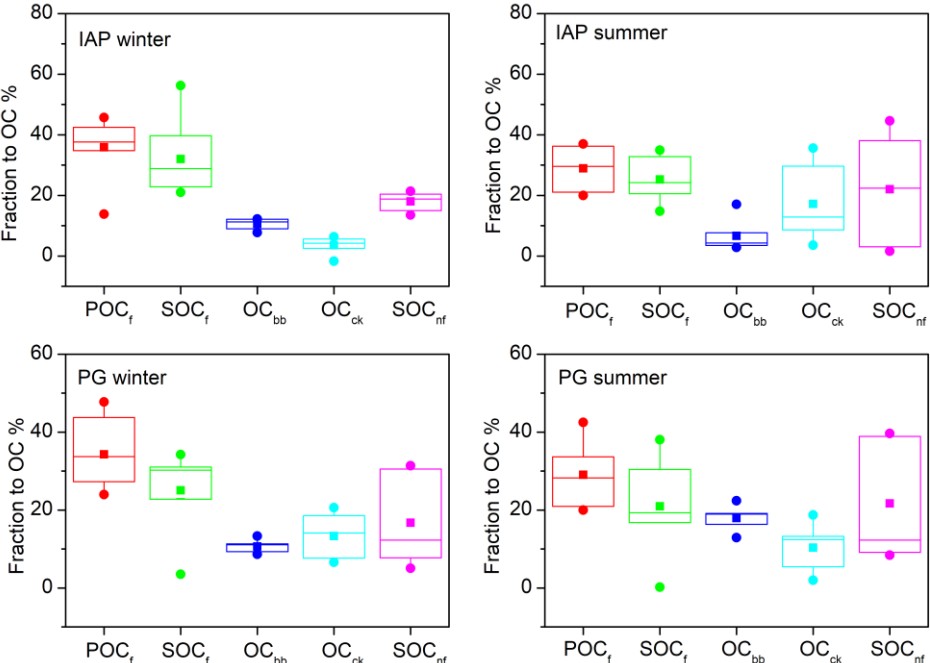

**Figure 6.** Fractions of each source (i.e., $POC_f$, $SOC_f$, $OC_{bb}$, $OC_{ck}$ and $SOC_{nf}$) in OC based on the extended Gelencsér method. f: fossil fuel sources, nf: non-fossil sources, bb: biomass burning, ck: cooking. The box denotes the 25th (lower line), 50th (middle line), and 75th (top line) percentiles; the solid squares within the box denote the mean values; the end of the vertical bars represents the 10th (below the box) and 90th (above the box) percentiles; and the solid dots denote maximum and minimum values.





**Figure 7.** Correlations of WINSOC, WSOC with POC, SOC at IAP and PG sites in winter and summer. The slopes and
correlation coefficients are summarized in Table S5.