# Peer review of "Source Apportionment of Carbonaceous Aerosols in Beijing with Radiocarbon and Organic Tracers: Insight into the Differences between Urban and Rural Sites"

_Atmospheric Chemistry and Physics, 2020_

## Referee Comment (RC1) · Anonymous Referee #1 · 21 Dec 2020

The title of this paper accurately describes its content. Measurements of the C14 content of 25 daily samples of PM2.5 collected at an urban site within Beijing and a rural site to the north of Beijing in both summer and winter were combined with OC, EC and organic tracer data to estimate contributions of different sources to the carbonaceous material within the PM2.5 samples. The method follows that applied to similar datasets of C14, OC-EC and tracer content of PM samples elsewhere. The work here extends the sources in the apportionment to include estimated contributions from cooking and a partitioning of biomass burning between wood and straw.

The general methodology requires certain assumptions about the number (and/or predominance) of certain contributing source types and on the generality of ratios of, for example, OC-to-EC or levoglucosan-to-OC, to yield the quantitative apportionment. The sensitivity of the conclusions of the quantification to these assumptions can be investigated using sensitivity calculations. In this work, the quantitative source apportionment was also compared against a chemical mass balance model. The two approaches agreed well aside from the cooking aerosol.

This work has been very carefully undertaken. The descriptions of the methods and of the interpretation of the data analyses are comprehensive and clear. The overall presentation quality of the manuscript is excellent. Likewise the depth and appropriateness of citation to previous work.

As well as providing novel insight into the source apportionment of the carbonaceous fraction at the sampling locations, the paper provides useful methodological approach for this sort of source apportionment.

I have a couple of general comments and only a few technical comments.

The first general observation is the really remarkably low spread (as represented by what is presumably a standard deviation) in the source proportion percentages, for example in the standard deviation of OCbb percentage for a given site and season, given the range in absolute PM2.5 concentrations. I might have expected that different sources would contribute proportionally greater or lesser to the carbonaceous PM2.5 as prevailing meteorology caused absolute PM2.5 concentrations vary. The discussion in the paper doesn't discuss within-season/site variability (or lack of) but only on between season/site variability.

The second general comment concerns statements in lines 152 and 156 that a value of 1.10 is applied to derive a fraction contemporary for EC under the assumption that the biomass burning is the only non-fossil source of EC. Whilst it is okay to assume that biomass burning is the only (or overwhelmingly dominant) non-fossil source of EC,
surely the appropriate correction factor depends on the age assumed for the biomass contributing to the EC?

**Technical comments**

L152: Should fM read fNF,ref here?

L158: Are the proportions stated here of 0.9 for biogenic and 0.1 for biomass burning the wrong way around?

L205: Use singular 'representative' here.

Figure 1 caption: The caption has 'left' and 'right' the wrong way around. Also add the respective years for the summer and winter dates.

Figure 2 caption: State the year for the summer and winter dates.

Figure 3 caption: Define LG, MN and GA in the caption. Also, correct 'LG/MA' to 'LG/GA'.

Figure 4 caption: change 'versus' to 'add'; the figure is not plotting one variable versus the second.

Figure 5: sort out the text that describes what is plotted in p

**ACPD**

---

## Referee Comment (RC2) · Anonymous Referee #2 · 19 Feb 2021

The authors incorporated the 14C analysis into the source apportionment method developed by Gelencsér et al. to apportion the fossil and non-fossil sources of EC and primary and secondary OC in Beijing. A total of 25 PM2.5 samples collected at an urban and a rural site in summer and winter were analyzed. An evident seasonal variation of fossil/non-fossil and primary/secondary OC was observed for both sites. Although the method used in this work is at the preliminary stage, it provides a new way to apportion primary and secondary OC from fossil and non-fossil sources. The relatively good agreements between the source apportionment results obtained from this

work and CMB models suggest the feasibility of this 14C analysis-incorporated Gelencsér method. The current model only includes levoglucosan as the source marker for biomass burning. Inclusion of more source markers in this method, such as hopanes for gasoline vehicles, cholesterol for cooking, and Al and Si for soil dust, may give a more explicit source apportionment of non-fossil and fossil-derived OC. Overall, the manuscript is well written, and I recommended it for publication with some comments provided below.

Major comments:

1. Previous studies on the source apportionment of PM2.5 in China showed that dust (including soil dust and road dust) is an important contributor to carbonaceous aerosols in Northern China (Huang et al., 2014; Zhang and Cao, 2015). Its contributions to both PM2.5 and organic matter in Beijing were even higher than that from the cooking source when PM2.5 mass concentrations ranged from 60 to 200 ug m-3 (Huang et al., 2014). So why contributions of dust to EC and OC were not counted in this study? One of the major assumptions made in this study is that all ECnf is from ECbb, which ignored the contribution from dust and may overestimate ECbb. Since OCbb was derived from ECbb, and the difference between POCnf and OCbb was 100% attributed to OCck, which overestimated the contributions from the cooking source. This is reflected by the poor correlation between OCck-this study and OCck-CMB. As shown in figure 5e, most of the OCck values derived in this study were larger than those apportioned by CMB, which further proved that other sources, e.g., dust, that contributed to OCnf and ECnf have been overlooked in this method. Si and Al are good markers for soil dust. Similar to what the authors did for OCbb, the authors may try to use the measured concentrations of Al and Si, and the ratios of these two markers to OC or EC in the dust source profiles to derived OCdust and ECdust. Similarly, cholesterol (data reported in Wu et al., 2020) and its ratios to OC and EC in cooking emission profiles can be used to derive OCcooking and ECcooking.

Zhang, Y.L. and Cao, F., Fine particulate matter (PM2.5) in China at a city level, Sci.

Rep., 5, 14884, 2015.

Huang et al., 2014 and Wu et al., 2020, see the references in the manuscript.

2. Some of the key parameters, such as the fractions of levoglucosan from softwood burning and straw burning, were empirically derived using these 25 samples; emission source profiles of softwood burning, straw burning, and maize burning were also empirically selected based on their best fit to the measured ECnf values of these 25 samples. These parameters and source profiles were then used to apportion the non-fossil and fossil-derived POC, SOC, and EC from the same batch of 25 PM2.5 samples. If possible, another set of Beijing PM2.5 samples could be analyzed using the parameters and source profiles determined in this study to validate the extended Gelencsér method and prove its general application to Beijing samples.

Minor comments:

1. Line 143-144: How much filter was extracted by water? By which extraction technique and for how long?

2. Line 147-148, How great will these factors influence the accuracy of the results? Which one is most crucial?

3. The authors mentioned several times (e.g., line 70 and 249) in the manuscript that coal combustion must be included in the extended Gelencsér method. However, in this study, coal combustion still cannot be explicitly differentiated from the fossil-derived OC and EC from vehicle emissions. How would the author improve this?

3. Since the focus is on the 25 selected PM2.5 samples in this study, it makes more sense to me to present the compound concentrations and meteorological parameters of just these 25 samples other than the whole batch of samples in Table 2.

4. The authors may think about moving Table 3 to the SI, since it is not critical for the discussion. 5. I suggest the authors include the time series of apportioned POCf, SOCf, OCbb, OCck, and SOCnf at IAP and PG in figure 6 as well. This will help the

readers to follow the discussion more easily.

---

## Author Comment (AC1) · 1 Apr 2021

The response to reviewers are detailed in the attached Supplement file.

Please also note the supplement to this comment:
https://acp.copernicus.org/preprints/acp-2020-1018/acp-2020-1018-AC1-supplement.pdf

---

## Author Response (AR1)

Title: Source Apportionment of Carbonaceous Aerosols in Beijing with Radiocarbon and Organic Tracers: Insight into the Differences between Urban and Rural Sites
Author(s): Siqi Hou et al.
MS No.: acp-2020-1018
Special Issue: In-depth study of air pollution sources and processes within Beijing and its surrounding region (APHH-Beijing) (ACP/AMT inter-journal SI)

**RESPONSE TO REVIEWERS**

**REVIEWR #1**
We are grateful to the Referee for the comments and the constructive suggestions to improve our manuscript. We have implemented all the comments and suggestions in the revised manuscript. Our point-to-point responses to the individual comment are as follows, we repeat the specific points raised by the reviewer in bold font, followed by our responses in italic font.

(1) The first general observation is the really remarkably low spread (as represented by what is presumably a standard deviation) in the source proportion percentages, for example in the standard deviation of OCbb percentage for a given site and season, given the range in absolute PM2.5 concentrations. I might have expected that different sources would contribute proportionally greater or lesser to the carbonaceous PM2.5 as prevailing meteorology caused absolute PM2.5 concentrations vary. The discussion in the paper doesn't discuss within-season/site variability (or lack of) but only on between season/site variability.
**RESPONSE:** Standard deviations of source contributions to OC derived by the extended Gelencsér method have been listed in the manuscript in line 306 and line 387. Here, we summarize the average contributions and their standard deviations in Table 1. As mentioned in line 132, the 25 samples for $^{14}C$ analysis were selected to represent different air quality conditions in winter and summer. We have amended the text by adding the following:

"The standard deviations appear small, but obscure the marked differences between seasons. Also, the time series in Fig. 6 (below) shows substantial day-to-day variations in the source contributions within a season, but still suggest that meteorological drivers play a major role in determining daily concentrations."

"22 November and 1 December at IAP and PG sites were lower than 75 µg m$^{-3}$ and regarded as non-haze air days, in contrast to other wintertime samples collected during haze pollution days." In summer, the average PM$_{2.5}$ concentrations of the whole campaign were 30.2 ± 14.8 µg m$^{-3}$ and 27.5 ± 12.9 µg m$^{-3}$ at the IAP and PG sites, while the PM$_{2.5}$ concentrations of samples for $^{14}C$ analysis were somewhat higher (42.5 ± 26.5 and 42.7 ± 21.2 µg m$^{-3}$ at IAP and PG). This is because we included two samples under influence of open burning (16 and 17 June) at both sites.

The summary of concentrations of PM$_{2.5}$, OC, EC, non-fossil fractions and meteorological parameters for the 25 selected samples can be found in revised Table 2 as below. We have also revised Figure 6 (shown as Figure 1) to show the time variations of the different source fractions of OC by the extended Gelencsér method.

As the referee suggests, we have added a discussion of variations of OC source proportions within season/site as follows:

"In the winter sampling campaign at IAP, POC$_f$ was the biggest contributor to OC followed by SOC$_f$. Both of them were significantly enhanced during haze periods, while the non-fossil fractions, OC$_{bb}$, OC$_{ck}$ and SOC$_{nf}$, did not show much difference between haze and non-haze periods. It implied the haze pollution at IAP in winter was elevated by the accumulation of coal combustion and traffic emissions under favourable weather conditions. The formation of secondary OC associated with coal combustion and traffic emissions was increased during the haze period.

In the winter campaign at PG, POC$_f$ and SOC$_f$ were the top two contributors to OC; however, the contribution of POC$_f$ and SOC$_f$ did not increase much in the haze period. In contrast, the fractions of SOC$_{nf}$ increased substantially on 3 and 4 December, on which days there were found to be open burning activities in surrounding areas (shown from the fire spots on Figure S2 in SI). It showed that a large proportion of OC$_{bb}$ was transformed to secondary OC during the transport of biomass burning aerosols to the receptor sites.

In summer, the sudden increase of the $SOC_{nf}$ fraction on 16 and 17 June at both sites was accompanied by an increase of $PM_{2.5}$ and OC concentrations. This is likely due to the open burning activities in surrounding areas. The enhancement of $(OC/EC)_{nf}$ ratios and $WSOC_{nf}$ fractions also suggested secondary OC formation through oxidation of primary non-fossil sources."

A detailed discussion of OC source apportionment between haze and non-haze days at IAP has already been published in another paper (Xu et al., 2021).

**Table 1** summary of source fractions of $OC_{bb}$, $OC_{ck}$, $POC_f$, $SOC_f$, and $SOC_{nf}$ to OC derived by the extended Gelencsér method in winter and summer at both sites (mean ± SD).

|  | IAP | | PG | |
|---|---|---|---|---|
|  | winter | summer | winter | summer |
| $OC_{bb}$ | 10.6±1.7% | 10.4±1.5% | 6.5±5.2% | 17.9±3.5% |
| $OC_{ck}$ | 3.6±2.7% | 13.4±5.8% | 17.4±12.5% | 10.4±6.7% |
| $POC_f$ | 35.8±10.5% | 34.1±8.7% | 28.9±7.4% | 29.1±9.4% |
| $SOC_f$ | 32.0±12.5% | 25.2±10.4% | 25.2±7.6% | 21.0±14.4% |
| $SOC_{nf}$ | 18.0±2.9% | 16.9±10.8% | 22.0±17.6% | 21.7±16.1% |

**Table 2** (shown as revised Table 2 in the manuscript) Statistical summary of concentrations, ratios and meteorological parameters at IAP and PG sites during winter and summer campaigns (mean ± SD).

| Compound/ Meteorological parameters | IAP | | | | | PG | | | | |
|---|---|---|---|---|---|---|---|---|---|---|
| | Winter | | | Summer | | Winter | | | Summer | |
| | All samples (n=32) | $^{14}$C samples (haze, n=5) | $^{14}$C samples (non-haze, n=2) | All samples (n=34) | $^{14}$C samples (n=6) | All samples (n=32) | $^{14}$C samples (haze, n=5) | $^{14}$C samples (non-haze, n=2) | All samples (n=34) | $^{14}$C samples (n=5) |
| $PM_{2.5}$($\mu$g m$^{-3}$) | 97.7±75.3 | 158.7±62.1 | 30.1±27.3 | 30.2±14.8 | 42.5±26.5 | 99.7±77.8 | 212.1±84.9 | 28.9±17.1 | 27.5±12.9 | 42.7±21.2 |
| OC($\mu$g m$^{-3}$) | 20.5±12.2 | 33.8±8.6 | 9.4±7.4 | 6.4±2.3 | 8.3±3.2 | 33.2±22.0 | 62.0±19.4 | 16.4±6.1 | 7.7±3.4 | 11.5±4.9 |
| EC($\mu$g m$^{-3}$) | 3.3±2.0 | 4.8±1.3 | 1.2±1.4 | 0.9±0.4 | 1.1±0.3 | 3.7±2.3 | 6.7±1.6 | 2.0±0.5 | 1.2±0.7 | 2.0±0.7 |
| OC/EC | 6.9±2.4 | 7.1±0.7 | 10.8±5.9 | 7.6±2.2 | 7.1±1.9 | 9.0±1.9 | 9.3±2.5 | 8.0±1.0 | 9.0±6.7 | 6.0±1.3 |
| LG(ng m$^{-3}$) | 310.7±196 | 431.6±160.3 | 144.2±68.1 | 27.9±29.6 | 49.7±65.3 | 634.3±483.2 | 1162.3±427.2 | 263.9±66.9 | 74.0±34.2 | 106.0±67.2 |
| MN(ng m$^{-3}$) | 32.4±20.7 | 44.0±16.8 | 14.3±4.9 | 2.6±2.4 | 4.1±5.0 | 81.1±63.2 | 146.1±37.3 | 29.3±0.1 | 10.3±3.7 | 12.8±5.0 |
| GA(ng m$^{-3}$) | 60.3±39.7 | 82.3±33.6 | 28.8±11.4 | 4.3±2.5 | 5.3±3.4 | 190.2±172.9 | 363.3±209.8 | 50.6±20.1 | 14.2±7.6 | 17.9±4.6 |
| $K^+$($\mu$g m$^{-3}$) | 1.2±1.0 | 1.6±0.8 | 0.3±0.2 | 0.4±0.4 | 0.7±0.7 | 1.6±1.3 | 2.8±1.3 | 0.4±0.2 | 0.5±0.3 | 0.8±0.5 |
| $f_{NF, EC}$ | na | 0.32±0.03 | 0.45±0.07 | na | 0.46±0.09 | na | 0.39±0.07 | 0.52±0.00 | na | 0.41±0.10 |
| $f_{NF, OC}$ | na | 0.32±0.05 | 0.32±0.03 | na | 0.46±0.12 | na | 0.40±0.07 | 0.42±0.02 | na | 0.50±0.09 |
| WS (m s$^{-1}$) | 2.7±1.1 | 2.0±0.4 | 3.1±1.0 | 3.6±0.8 | 4.2±1.2 | 1.4±0.8 | 0.9±0.2 | 2.0±1.8 | 1.1±0.6 | 0.3±0.2 |
| RH (%) | 41.8±15.1 | 44.9±8.0 | 27.6±3.9 | 43.7±16.6 | 32.0±12.1 | 51.7±13.2 | 73.8±11.7 | 47.3±8.0 | 50.2±14.3 | 39.9±4.3 |
| T (ºC) | 7.0±3.3 | 7.0±1.4 | 4.7±4.9 | 24.6±3.6 | 27.1±3.7 | 1.0±3.2 | 0.7±1.9 | -1.5±4.7 | 23.3±3.7 | 26.9±3.2 |

[Figure]

**Figure 1** (shown as revised Figure 6 in the manuscript) Time variations of OC source apportionment results by extended Gelencsér method (upper) the fractions of each source (i.e., $POC_f$, $SOC_f$, $OC_{bb}$, $OC_{ck}$ and $SOC_{nf}$) in OC based on the extended Gelencsér method (lower). f: fossil fuel sources, nf: non-fossil sources, bb: biomass burning, ck: cooking. The box denotes the 25th (lower line), 50th (middle line), and 75th (top line) percentiles; the solid squares within the box denote the mean values; the end of the vertical bars represents the 10th (below the box) and 90th (above the box) percentiles; and the solid dots denote maximum and minimum values.

(2) The second general comment concerns statements in lines 152 and 156 that a value of 1.10 is applied to derive a fraction contemporary for EC under the assumption that the biomass burning is the only non-fossil source of EC. Whilst it is okay to assume that biomass burning is the only (or overwhelmingly dominant) non-fossil source of EC, surely the appropriate correction factor depends on the age assumed for the biomass contributing to the EC?

**RESPONSE**: The correction factor is affected by the age, weighting function and relative share of different biomass materials. Lewis et al. (2004) showed that for the $^{14}C$ content of trees aged up to 75 years at the harvest time in 1999, the correction factor was at least 1.08 and no more than 1.25. The factor used in this study (1.10 ± 0.05) is derived by a tree growth model with low and high limits of 1.05 and 1.15 considering ages, weights and relative shares (Mohn et al., 2008).

(3) Technical comments

L152: Should fM read fNF,ref here?

**RESPONSE**: *Right, the $f_M$ has been revised to $f_{NF,ref}$.*

L158: Are the proportions stated here of 0.9 for biogenic and 0.1 for biomass burning the wrong way around?

**RESPONSE:** Yes, the sentence should be revised to "while $p_{bio}$ and $p_{bb}$ are the proportions of biogenic source and biomass burning respectively, which are 0.1 and 0.9 in winter and 0.5 and 0.5 in summer".

L205: Use singular 'representative' here.

**RESPONSE:** The "representatives" has been revised to "representative".

Figure 1 caption: The caption has 'left' and 'right' the wrong way around. Also add the respective years for the summer and winter dates.

**RESPONSE:** The left and right has been exchanged, and years are added. The caption has been revised to "Time series of $PM_{2.5}$ and its major components at IAP and PG during winter in 2016 (left) and summer in 2017 (right)."

Figure 2 caption: State the year for the summer and winter dates.
**RESPONSE:** The caption has been revised to "Time series of concentrations of $WSOC_f$, $WSOC_{nf}$, $WINSOC_f$, $WINSOC_{nf}$, $EC_f$ and $EC_{nf}$ (left) and their relative contributions to TC (right) during winter in 2016 and summer in 2017 at IAP and PG."

Figure 3 caption: Define LG, MN and GA in the caption. Also, correct 'LG/MA' to 'LG/GA'.
**RESPONSE:** Added "LG, MN and GA refer to levoglucosan, mannosan and galactosan, respectively" Revised "LG/MA" to "LG/MN". Also, revised the LG/MA in line 282.

Figure 4 caption: change 'versus' to 'add'; the figure is not plotting one variable versus the second.
**RESPONSE:** The caption has been revised to "Concentrations of OC from softwood ($OC_{wood}$) and OC from straw ($OC_{straw}$) at IAP (upper) and PG (lower) and variations of $OC_{wood}$ fractions."

Figure 5: sort out the text that describes what is plotted in p
**RESPONSE:** Added descriptions "Good correlations were found for the apportioned $POC_f$, $POC_{nf}$, SOC and $OC_{bb}$ between the CMB and EG methods, while there were large discrepancies in $OC_{ck}$. The PMF model did not show separate source apportionment results from fossil, non-fossil or secondary OC, so only $OC_{ck}$ concentrations were compared."

**References:**
Lewis, C. W., Klouda, G. A., and Ellenson, W. D.: Radiocarbon measurement of the biogenic contribution to summertime PM2.5 ambient aerosol in Nashville, TN, Atmos. Environ., 38, 6053-6061, https://10.1016/j.atmosenv.2004.06.011, 2004.

Mohn, J., Szidat, S., Fellner, J., Rechberger, H., Quartier, R., Buchmann, B., and Emmenegger, L.: Determination of biogenic and fossil $CO_2$ emitted by waste incineration based on $14CO_2$ and mass balances, Bioresource Technology, 99, 6471-6479, https://doi.org/10.1016/j.biortech.2007.11.042, 2008.

Xu, J., Srivastava, D., Wu, X., Hou, S., Vu, Tuan V., Liu, D., Sun, Y., Vlachou, A., Moschos, V., Salazar, G., Szidat, S., Prévôt, A. S. H., Fu, P., Harrison, R. M., and Shi, Z.: An evaluation of source apportionment of fine OC and PM2.5 by multiple methods: APHH-Beijing campaigns as a case study, Faraday Discussions, https://doi.org/10.1039/D0FD00095G, 2021.

**REVIEWER #2**
We are grateful to the Referee for the comments and the constructive suggestions to improve our manuscript. We have addressed all the comments and suggestions in the revised manuscript. Our point-to-point responses to the individual comment are as follows, we repeat the specific points raised by the reviewer in bold font, followed by our responses in italic font.

(1) Previous studies on the source apportionment of PM2.5 in China showed that dust (including soil dust and road dust) is an important contributor to carbonaceous aerosols in Northern China (Huang et al., 2014; Zhang and Cao, 2015). Its contributions to both PM2.5 and organic matter in Beijing were even higher than that from the cooking source when PM2.5 mass concentrations ranged from 60 to 200 ug m-3 (Huang et al., 2014). So why contributions of dust to EC and OC were not counted in this study? One of the major assumptions made in this study is that all $EC_{nf}$ is from $EC_{bb}$, which ignored the contribution from dust and may overestimate $EC_{bb}$. Since $OC_{bb}$ was derived from $EC_{bb}$, and the difference between $POC_{nf}$ and $OC_{bb}$ was 100% attributed to $OC_{ck}$, which overestimated the contributions from the cooking source. This is reflected by the poor correlation between $OC_{ck}$-this study and $OC_{ck}$-CMB. As shown in figure 5e, most of the $OC_{ck}$ values derived in this study were larger than those apportioned by CMB, which further proved that other sources, e.g., dust, that contributed to $OC_{nf}$ and $EC_{nf}$ have been overlooked in this method. Si and Al are good markers for soil dust. Similar to what the authors did for $OC_{bb}$, the authors may try to use the measured concentrations of Al and Si, and the ratios of these two markers to OC or EC in the dust source profiles to derived $OC_{dust}$ and $EC_{dust}$. Similarly, cholesterol (data reported in Wu et al., 2020) and its ratios to OC and EC in cooking emission profiles can be used to derive $OC_{cooking}$ and $EC_{cooking}$.

Zhang, Y.L. and Cao, F., Fine particulate matter (PM2.5) in China at a city level, Sci. Rep., 5, 14884, 2015.
Huang et al., 2014 and Wu et al., 2020, see the references in the manuscript.

**RESPONSE:** As the referee suggests, we calculated the OC from cooking by cholesterol concentrations multiplying OC to cholesterol ratios (Zhao et al., 2015; Wu et al., 2021), and added discussion as follow in the SI:

"The concentrations of cholesterol in 25 selected samples with the corresponding OC from cooking ($OC_{ck-ch}$) are summarized in Table S5. The methodology of cholesterol determination is described in Xu et. al (2020). The average concentrations of $OC_{ck-ch}$ are 2.08±1.16 μg m$^{-3}$ and 1.64±1.01 μg m$^{-3}$ at IAP in winter and summer, 2.65 ±1.06 μg m$^{-3}$ and 0.92±0.43 μg m$^{-3}$ at PG in winter and summer. The $OC_{ck-ch}$ concentrations are 1.8 times higher than the $OC_{ck}$ from the EG method ($OC_{ck-EG}$) on average at IAP, and will result in the values of $OC_{bb}$ + $OC_{ck-ch}$ being much higher than $POC_{nf}$. It is suggested that the $OC_{ck-ch}$ may contain some secondary OC. At PG, however, concentrations of $OC_{ck-ch}$ are only half of $OC_{ck-EG}$. the $OC_{ck-EG}$ is calculated by subtracting $OC_{bb}$ from $POC_{nf}$ assuming it arises mainly from cooking. Here, the much higher $OC_{ck-EG}$ than $OC_{ck-ch}$ at PG suggest that the $OC_{ck-EG}$ may include other primary sources.

Comparisons of $OC_{ck-ch}$ with $OC_{ck-EG}$, $OC_{ck}$ from the CMB model and cooking OC from AMS/ACSM-PMF are shown in Figure 1(Figure S8 in SI). The concentrations of $OC_{ck-ch}$ are not well correlated with CMB results or AMS/ACSM-PMF results. But the $OC_{ck-ch}$ values are 6.5 times higher than CMB results on average, and 0.91 times the AMS/ACSM-PMF results. It is possible that the $OC_{ck-ch}$ may contain secondary OC.

$OC_{ck-ch}$ at PG is half of $OC_{ck-EG}$. We found the differences between $OC_{ck-EG}$ and $OC_{ck-ch}$ ($OC_{onf}$) at PG are positively correlated with crustal elements, Si, Al, Fe and Ti (shown in Figure 2, Figure S9 in SI). This indicates that the $OC_{ck-EG}$ may include OC fractions from primary sources like dust. The filters collected during the APHH-campaign have been subject to elemental analysis with XRF and ICP-MS. The detailed methods of elemental analysis can be found in Srivastava et al (2020).

**Table 1** (shown as Table S5 in SI) summary of Cholesterol and element concentrations, EFs, $OC_{ck-ch}$, $OC_{onf}$, $OC_{dt}$ and $OC_{dt-Al}$.

| Site | Date | Cholesterol ng m$^{-3}$ | Si ng m$^{-3}$ | Al ng m$^{-3}$ | Fe ng m$^{-3}$ | Ti ng m$^{-3}$ | $OC_{ck-ch}$ µg m$^{-3}$ | $OC_{onf}$ µg m$^{-3}$ | EF(Si) | EF(Fe) | EF (Ti) | $OC_{dt}$ µg m$^{-3}$ | $OC_{dt-Al}$ µg m$^{-3}$ |
|------|------|------|------|------|------|------|------|------|------|------|------|------|------|
| IAP | 22/11/2016 | 1.10 | 141.6 | 68.8 | 190.4 | 34.6 | 2.16 | -2.23 | 0.71 | 2.64 | 6.38 | 0.02-0.14 | 0.03-0.15 |
| IAP | 24/11/2016 | 1.39 | 335.2 | 229.5 | 526.9 | 8.8 | 2.71 | -1.80 | 0.50 | 2.19 | 0.49 | 0.05-0.32 | 0.10-0.50 |
| IAP | 26/11/2016 | 1.08 | 2819.5 | 1372.1 | 1053.1 | 68.8 | 2.11 | -0.11 | 0.71 | 0.73 | 0.64 | 0.41-2.70 | 0.58-3.01 |
| IAP | 01/12/2016 | 1.61 | 459.7 | 313.7 | 350.7 | 0.0 | 3.15 | -2.22 | 0.51 | 1.06 | 0.00 | 0.07-0.44 | 0.13-0.69 |
| IAP | 02/12/2016 | 0.18 | 778.7 | 569.8 | 435.8 | 3.1 | 0.36 | 0.62 | 0.47 | 0.73 | 0.07 | 0.11-0.75 | 0.24-1.25 |
| IAP | 03/12/2016 | 0.37 | 1551.3 | 1319.9 | 1032.8 | 25.6 | 0.72 | 1.17 | 0.41 | 0.75 | 0.25 | 0.22-1.49 | 0.55-2.90 |
| IAP | 04/12/2016 | 1.73 | 2244.7 | 1635.2 | 467.0 | 18.3 | 3.38 | -2.34 | 0.47 | 0.27 | 0.14 | 0.33-2.15 | 0.69-3.59 |
| IAP | 24/05/2017 | 0.46 | 43.4 | 25.0 | 283.9 | 5.7 | 0.90 | 0.77 | 0.60 | 10.82 | 2.89 | 0.01-0.04 | 0.01-0.05 |
| IAP | 26/05/2017 | 1.14 | 645.2 | 297.3 | 621.8 | 27.6 | 2.22 | -1.05 | 0.75 | 1.99 | 1.18 | 0.09-0.62 | 0.12-0.65 |
| IAP | 27/05/2017 | 1.65 | 741.6 | 346.6 | 1149.8 | 32.1 | 3.22 | -1.83 | 0.74 | 3.16 | 1.17 | 0.11-0.71 | 0.15-0.76 |
| IAP | 10/06/2017 | 0.38 | 1102.6 | 466.2 | 579.6 | 32.7 | 0.73 | 0.64 | 0.82 | 1.18 | 0.89 | 0.16-1.06 | 0.20-1.02 |
| IAP | 16/06/2017 | 0.40 | 793.0 | 302.9 | 433.0 | 37.3 | 0.78 | -0.05 | 0.90 | 1.36 | 1.56 | 0.11-0.76 | 0.13-0.67 |
| IAP | 17/06/2017 | 1.00 | 584.0 | 188.2 | 488.7 | 19.9 | 1.96 | -1.44 | 1.07 | 2.47 | 1.34 | 0.08-0.56 | 0.08-0.41 |
| PG | 22/11/2016 | 0.70 | n.a | 55.8 | 220.2 | 18.9 | 1.36 | 1.14 | n.a | 3.76 | 4.30 | n.a | 0.02-0.12 |
| PG | 24/11/2016 | 1.25 | n.a | 395.8 | 1354.9 | 51.9 | 2.44 | 4.31 | n.a | 3.26 | 1.66 | n.a | 0.17-0.87 |
| PG | 26/11/2016 | 1.75 | n.a | 1153.9 | 1979.3 | 164.7 | 3.42 | 9.65 | n.a | 1.63 | 1.81 | n.a | 0.48-2.53 |
| PG | 01/12/2016 | 1.08 | n.a | 111.2 | 244.5 | 2.6 | 2.11 | 1.45 | n.a | 2.09 | 0.30 | n.a | 0.05-0.24 |
| PG | 02/12/2016 | 0.94 | n.a | 452.6 | 625.0 | 5.6 | 1.84 | 1.03 | n.a | 1.32 | 0.16 | n.a | 0.19-0.99 |
| PG | 03/12/2016 | 1.46 | n.a | 897.2 | 1206.4 | 32.5 | 2.85 | 3.17 | n.a | 1.28 | 0.46 | n.a | 0.38-1.97 |

| | | | | | | | | | | | | |
|---|---|---|---|---|---|---|---|---|---|---|---|---|
| PG | 04/12/2016 | 2.30 | n.a | 322.1 | 381.3 | 7.0 | 4.50 | 1.14 | n.a | 1.13 | 0.28 | n.a | 0.14-0.71 |
| PG | 26/05/2017 | 0.31 | 480.0 | 273.3 | 9.5 | 29.4 | 0.60 | 0.47 | 0.61 | 0.03 | 1.36 | 0.07-0.46 | 0.11-0.60 |
| PG | 27/05/2017 | 0.17 | 736.9 | 338.0 | 17.5 | 29.2 | 0.34 | 1.87 | 0.75 | 0.05 | 1.10 | 0.11-0.71 | 0.14-0.74 |
| PG | 10/06/2017 | 0.65 | 614.9 | 257.1 | 4.9 | 24.7 | 1.26 | 0.31 | 0.82 | 0.02 | 1.22 | 0.09-0.59 | 0.11-0.56 |
| PG | 16/06/2017 | 0.68 | 786.8 | 325.3 | 9.2 | 27.2 | 1.32 | 5.92 | 0.83 | 0.03 | 1.06 | 0.11-0.75 | 0.14-0.71 |
| PG | 17/06/2017 | 0.55 | 674.3 | 217.7 | 10.9 | 23.3 | 1.07 | 0.57 | 1.07 | 0.05 | 1.36 | 0.10-0.65 | 0.09-0.48 |

[Figure]

**Figure 1** (shown as Figure S8 in SI) Correlations of OCck-ch with OCck-EG, OCck-CMB and COC-AMS/ACSM-PMF. OCck-ch, OC from cooking from cholesterol concentrations and cholesterol to OC ratios; OCck-EG, OC from cooking from extended Gelencsér method; OCck-CMB, OC from cooking from CMB model; COC-AMS/ACSM-PMF, OC from cooking from AMS/ACSM-PMF model (AMS for IAP and ACSM for PG).

[Figure]

**Figure 2** (shown as Figure S9 in SI) Correlations of OC$_{onf}$ (=OC$_{ck-EG}$-OC$_{ck-ch}$) with Si (no data in winter campaign of PG), Al, Fe and Ti. OC$_{ck-ch}$, OC from cooking from cholesterol concentrations and cholesterol to OC ratios; OC$_{ck-EG}$, OC from cooking from extended Gelencsér method.

Enrichment factors (EFs) can be used to study the degree of elemental enrichment in ambient particles and can also help to determine whether they are from natural or anthropogenic emissions. The calculation of EFs are as follow,

$$EF = \frac{\left(C_x / C_{Al}\right)_{PM_{2.5}}}{\left(C_x / C_{Al}\right)_{Soil}}$$

Where, $\left(C_x/C_{Al}\right)_{PM_{2.5}}$ is the concentration ratio of x to Al in the measured PM2.5 samples, $\left(C_x/C_{Al}\right)_{Soil}$ is the concentration ratio of x to Al of fugitive dust in Chinese Loess Plateau (Cao et al., 2008), respectively. Here, Al is the reference element due to its stability and immunity to human interference (Uematsu et al., 1983; Zhang et al., 2003).

The EFs of Si, Fe and Ti are listed in Table 1 (shown as Table S5 in SI). EF(Si) is in the range of 0.41 to 1.07, indicating that Si is mostly from natural sources. EF(Fe) and EF(Ti) are in range of 0.02-10.82 and 0-6.38, respectively, indicating that Fe and Ti are from mixed sources. Thus, we used Si concentrations and the Si to OC ratio from the Chinese Loess Plateau (Cao et al., 2008) and from Beijing road dust samples (Hu et al., 2019) to calculate a possible range OC from dust (OC$_{dt}$). We also calculate OC from dust (OC$_{dt-Al}$) using Al concentrations and the Al to OC ratio for comparison. The ranges of OC$_{dt}$ and OC$_{dt-Al}$ are listed in Table 1. The OC$_{dt}$ and OC$_{dt-Al}$ would result in a contribution to OC of 0.1-22.8% and 0.2-22.1%, respectively. And the calculated OC$_{dt}$ would contribute 1.9% to 192.5% of OC$_{onf}$ for PG site. It implies the OC from dust may be a major contributor to the primary non-fossil sources at PG.

Our other research on source apportionment of PM2.5 using PMF has presented a detailed study of dust contributions (Srivastava et al., 2020). It showed that the crustal dust made a significant contribution to OC and EC. But it cannot clearly be attributed to soil dust or road dust, and contains mixed characteristics. The estimated dust contributions in urban Beijing were 12.7% during haze periods (PM2.5 > 75 μg m$^{-3}$) and 35.2% during non-haze periods (PM2.5 < 75 μg m$^{-3}$). The huge discrepancy between the methods is not easily explained, but Srivastava et al. (2020) urge caution in accepting their results.

(2) Some of the key parameters, such as the fractions of levoglucosan from softwood burning and straw burning, were empirically derived using these 25 samples; emission source profiles of softwood burning, straw burning, and maize burning were also empirically selected based on their best fit to the measured ECnf values of these 25 samples. These parameters and source profiles were then used to apportion the non-fossil and fossil-derived POC, SOC, and EC from the same batch of 25 PM2.5 samples. If possible, another set of Beijing PM2.5 samples could be analyzed using the parameters and source profiles determined in this study to validate the extended Gelencsér method and prove its general application to Beijing samples.
**RESPONSE:** It is a great suggestion to apply the EG method to other Beijing samples. but it is not possible to achieve now as we only analysed 25 samples for $^{14}$C during the APHH campaign. However, we can apply the EG method with $^{14}$C analysis to more samples in the future.

(3)Minor comments
Line 143-144: How much filter was extracted by water? By which extraction technique and for how long?
**RESPONSE**: For EC separation, filter samples were first extracted in water to minimize positive artifacts from OC charring. "Under a laminar flow box, 23-mm diameter discs are punched out of the filters, sandwiched between two sealing rings and placed with the laden side upwards on a 25-mm-diameter plastic filter holder (Sartorius GmbH, Germany) and topped by a plastic syringe body. 20 mL ultrapure water with low TOC impurity is then passed through the filter without a pump. The filter punch is then delicately removed and placed for several hours in the desiccator for drying. Finally, a 1.5 cm$^2$ rectangle is punched out of the water-extracted filter, wrapped in aluminium foil, packed into a sealed plastic bag and stored in the freezer (-18 $^o$C) until OC/EC analysis." (Zhang et al., 2012; Liu et al., 2014)

Line 147-148, How great will these factors influence the accuracy of the results? Which one is most crucial?
**RESPONSE:** There are several factors may affect the accuracy of f$_{NF}$ (non-fossil fractions).

For EC, the f$_{NF,EC}$ is charring corrected first and then divided by a reference value 1.10 ± 0.05, assuming biomass burning is the only source of non-fossil EC. The uncertainty of the charring correction is around 10%, and the value 1.10 ± 0.05 may also bring uncertainties during the calculations.

For OC, the non-fossil fraction (f$_{NF,OC}$) is divided by a reference (equation 2 in the article). The selection of proportions of biogenic source and biomass burning are referred according to Levin et al., 2010, which are 0.9 and 0.1 in winter and 0.5 and 0.5 in summer, and this can also bring uncertainties when deriving the non-fossil fractions of OC. The different proportions could result in a maximum uncertainty of 7%.

For these factors, the charring correction is the most crucial step that can lead to large bias on f$_{NF,EC}$, Therefore, charring should be reduced to a minimum for an optimised EC isolation for $^{14}$C analysis. The pretreatment of EC can guarantee a minimum charring uncertainty because the suppression of charring is especially achieved by water-extraction treatment on the one hand and oxidative treatment (i.e. combustion in pure $O_2$) of the filters on the other hand. The water-extraction treatment prior to the EC collection substantially reduces charring due to the removal of WSOC as well as of some inorganic catalytic compounds. Furthermore, charring is substantially smaller if pure oxygen is used for the OC step instead of helium.

The authors mentioned several times (e.g., line 70 and 249) in the manuscript that coal combustion must be included in the extended Gelencsér method. However, in this study, coal combustion still cannot be explicitly differentiated from the fossil-derived OC and EC from vehicle emissions. How would the author improve this?

**RESPONSE:** This is one of the weaknesses of our method. The separate fractions from coal combustions and traffic emissions can be estimated if we can get specific OC/EC ratios from coal combustion and traffic emissions. We have tried to calculate these fractions according to OC/EC ratios from previous studies. The results showed that compared to traffic emissions ($POC_{tr}$), coal combustion ($POC_{cc}$) is less important. The accuracy of the estimation of $POC_{cc}$ and $POC_{tr}$ is limited by the OC/EC ratios, which vary according to combustion conditions, fuel types and even measurement method. Thus, we only give a rough estimation in this article. With more specific local emission source profiles, the accuracy of the EG method can be improved in the future.

We have added the following text:

"Ni et al. (2018, 2019) reported $\delta^{13}C$ signatures of biomass burning, coal combustion and traffic emissions as well as the OC/EC ratios from previous literature. By combining stable carbon isotopic composition analysis of EC with $^{14}C$ analysis, the proportions of coal combustion and traffic emission to EC can be derived using Bayesian statistics. The introduction of stable carbon isotopic analysis is suggested as a way to improve our EG method."

Since the focus is on the 25 selected PM2.5 samples in this study, it makes more sense to me to present the compound concentrations and meteorological parameters of just these 25 samples other than the whole batch of samples in Table 2.

**RESPONSE:** This suggestion is really helpful and we have modified Table 2 and merged it with Table S1 as follows:

**Table 2** (shown as revised Table 2 in the manuscript) Statistical summary of concentrations, ratios and meteorological parameters at IAP and PG sites during winter campaigns

| Compound/ Meteorological parameters | IAP | | | | | PG | | | | |
|---|---|---|---|---|---|---|---|---|---|---|
| | Winter | | | Summer | | Winter | | | Summer | |
| | All samples (n=32) | $^{14}C$ samples (haze, n=5) | $^{14}C$ samples (non-haze, n=2) | All samples (n=34) | $^{14}C$ samples (n=6) | All samples (n=32) | $^{14}C$ samples (haze, n=5) | $^{14}C$ samples (non-haze, n=2) | All samples (n=34) | $^{14}C$ samples (n=5) |
| $PM_{2.5}$($\mu g\ m^{-3}$) | 97.7±75.3 | 158.7±62.1 | 30.1±27.3 | 30.2±14.8 | 42.5±26.5 | 99.7±77.8 | 212.1±84.9 | 28.9±17.1 | 27.5±12.9 | 42.7±21.2 |
| OC($\mu g\ m^{-3}$) | 20.5±12.2 | 33.8±8.6 | 9.4±7.4 | 6.4±2.3 | 8.3±3.2 | 33.2±22.0 | 62.0±19.4 | 16.4±6.1 | 7.7±3.4 | 11.5±4.9 |
| EC($\mu g\ m^{-3}$) | 3.3±2.0 | 4.8±1.3 | 1.2±1.4 | 0.9±0.4 | 1.1±0.3 | 3.7±2.3 | 6.7±1.6 | 2.0±0.5 | 1.2±0.7 | 2.0±0.7 |
| OC/EC | 6.9±2.4 | 7.1±0.7 | 10.8±5.9 | 7.6±2.2 | 7.1±1.9 | 9.0±1.9 | 9.3±2.5 | 8.0±1.0 | 9.0±6.7 | 6.0±1.3 |
| LG(ng $m^{-3}$) | 310.7±196 | 431.6±160.3 | 144.2±68.1 | 27.9±29.6 | 49.7±65.3 | 634.3±483.2 | 1162.3±427.2 | 263.9±66.9 | 74.0±34.2 | 106.0±67.2 |
| MN(ng $m^{-3}$) | 32.4±20.7 | 44.0±16.8 | 14.3±4.9 | 2.6±2.4 | 4.1±5.0 | 81.1±63.2 | 146.1±37.3 | 29.3±0.1 | 10.3±3.7 | 12.8±5.0 |
| GA(ng $m^{-3}$) | 60.3±39.7 | 82.3±33.6 | 28.8±11.4 | 4.3±2.5 | 5.3±3.4 | 190.2±172.9 | 363.3±209.8 | 50.6±20.1 | 14.2±7.6 | 17.9±4.6 |
| $K^{+}$($\mu g\ m^{-3}$) | 1.2±1.0 | 1.6±0.8 | 0.3±0.2 | 0.4±0.4 | 0.7±0.7 | 1.6±1.3 | 2.8±1.3 | 0.4±0.2 | 0.5±0.3 | 0.8±0.5 |
| $f_{NF,\ EC}$ | na | 0.32±0.03 | 0.45±0.07 | na | 0.46±0.09 | na | 0.39±0.07 | 0.52±0.00 | na | 0.41±0.10 |
| $f_{NF,\ OC}$ | na | 0.32±0.05 | 0.32±0.03 | na | 0.46±0.12 | na | 0.40±0.07 | 0.42±0.02 | na | 0.50±0.09 |
| WS (m $s^{-1}$) | 2.7±1.1 | 2.0±0.4 | 3.1±1.0 | 3.6±0.8 | 4.2±1.2 | 1.4±0.8 | 0.9±0.2 | 2.0±1.8 | 1.1±0.6 | 0.3±0.2 |
| RH (%) | 41.8±15.1 | 44.9±8.0 | 27.6±3.9 | 43.7±16.6 | 32.0±12.1 | 51.7±13.2 | 73.8±11.7 | 47.3±8.0 | 50.2±14.3 | 39.9±4.3 |
| T (°C) | 7.0±3.3 | 7.0±1.4 | 4.7±4.9 | 24.6±3.6 | 27.1±3.7 | 1.0±3.2 | 0.7±1.9 | -1.5±4.7 | 23.3±3.7 | 26.9±3.2 |

The authors may think about moving Table 3 to the SI, since it is not critical for the discussion.
**RESPONSE**: We agree and have moved Table 3 to SI as Table S1.

I suggest the authors include the time series of apportioned POCf, SOCf, OCbb, OCck, and SOCnf at IAP and PG in figure 6 as well. This will help the readers to follow the discussion more easily.
RESPONSE: This suggestion is really helpful and we have changed the figure as follow,

[Figure]

**Figure 3** (shown as revised Figure 6 in the manuscript) Time variations of OC source apportionment results by extended Gelencsér method (upper) and the fractions of each source (i.e., POCf, SOCf, OCbb, OCck and SOCnf) in OC based on the extended Gelencsér method (lower). f: fossil fuel sources, nf: non-fossil sources, bb: biomass burning, ck: cooking. The box denotes the 25th (lower line), 50th (middle line), and 75th (top line) percentiles; the solid squares within the box denote the mean values; the end of the vertical bars represents the 10th (below the box) and 90th (above the box) percentiles; and the solid dots denote maximum and minimum values.

**Reference:**
Agrios, K., Salazar, G., Zhang, Y.-L., Uglietti, C., Battaglia, M., Luginbühl, M., Ciobanu, V. G., Vonwiller, M., Szidat, S.: Online coupling of pure $O_2$ thermo-optical methods – $^{14}C$ AMS for source apportionment of carbonaceous aerosols, Nucl. Instrum. Methods Phys. Res., B, 361, 288-293, https://doi.org/10.1016/j.nimb.2015.06.008, 2015.

Cao, J. J., Chow, J. C., Watson, J. G., Wu, F., Han, Y. M., Jin, Z. D., Shen, Z. X., and An, Z. S.: Size-differentiated source profiles for fugitive dust in the Chinese Loess Plateau, Atmos. Environ., 42, 2261-2275, https://doi.org/10.1016/j.atmosenv.2007.12.041, 2008.

Hu, Y., Li, M., Yan, X., Zhang, C.: Characteristics and Interannual Variation of Chemical Components in Typical Road Dust in Beijing. Environmental Science, 40, 1645-1655, https://doi.org/10.13227/j.hjkx.201808224, 2019.

Liu, J., Li, J., Zhang, Y., Liu, D., Ding, P., Shen, C., Shen, K., He, Q., Ding, X., Wang, X., Chen, D., Szidat, S., and Zhang, G.: Source Apportionment Using Radiocarbon and Organic Tracers for PM2.5 Carbonaceous Aerosols in Guangzhou, South China: Contrasting Local- and Regional-Scale Haze Events, Environ. Sci. Technol., 48, 12002-12011, https://doi.org/10.1021/es503102w, 2014.

Ni, H. Y., Huang, R. J., Cao, J. J., Liu, W. G., Zhang, T., Wang, M., Meijer, H. A. J., and Dusek, U.: Source apportionment of carbonaceous aerosols in Xi'an, China: insights from a full year of measurements of radiocarbon and the stable isotope C-13, Atmospheric Chemistry and Physics, 18, 16363-16383, 10.5194/acp-18-16363-2018, 2018.

Ni, H. Y., Huang, R. J., Cao, J. J., Dai, W. T., Zhou, J. M., Deng, H. Y., Aerts-Bijma, A., Meijer, H. A. J., and Dusek, U.: High contributions of fossil sources to more volatile organic aerosol, Atmospheric Chemistry and Physics, 19, 10405-10422, 10.5194/acp-19-10405-2019, 2019.

Srivastava, D., Xu, J., Vu, T. V., Liu, D., Li, L., Fu, P., Hou, S., Shi, Z., Harrison, R. M.: Insight into PM2.5 sources by applying Positive Matrix factorization (PMF) at an urban and rural site of Beijing, (in review), 2020.

Uematsu, M., Duce, R. A., Prospero, J. M., Chen, L., Merrill, J. T., and McDonald, R. L.: Transport of mineral aerosol from Asia Over the North Pacific Ocean, Journal of Geophysical Research: Oceans, 88, 5343-5352, https://doi.org/10.1029/JC088iC09p05343, 1983.

Xu, J., Srivastava, D., Wu, X., Hou, S., Vu, Tuan V., Liu, D., Sun, Y., Vlachou, A., Moschos, V., Salazar, G., Szidat, S., Prévôt, A. S. H., Fu, P., Harrison, R. M., and Shi, Z.: An evaluation of source apportionment of fine OC and PM2.5 by multiple methods: APHH-Beijing campaigns as a case study, Faraday Discussions, https://doi.org/10.1039/D0FD00095G, 2021.

Zhang, X. Y., Gong, S. L., Shen, Z. X., Mei, F. M., Xi, X. X., Liu, L. C., Zhou, Z. J., Wang, D., Wang, Y. Q., and Cheng, Y.: Characterization of soil dust aerosol in China and its transport and distribution during 2001 ACE-Asia: 1. Network observations, Journal of Geophysical Research: Atmospheres, 108, https://doi.org/10.1029/2002JD002632, 2003.

Zhang, Y. L., Perron, N., Ciobanu, V. G., Zotter, P., Minguillón, M. C., Wacker, L., Prévôt, A. S. H., Baltensperger, U., and Szidat, S.: On the isolation of OC and EC and the optimal strategy of radiocarbon-based source apportionment of carbonaceous aerosols, Atmos. Chem. Phys, 12, 10,841-10,856, https://doi.org/10.5194/acp-12-10841-2012, 2012.

Zhao, X. Y., Hu, Q. H., Wang, X. M., Ding, X., He, Q. F., Zhang, Z., Shen, R. Q., Lu, S. J., Liu, T. Y., Fu, X. X., and Chen, L. G.: Composition profiles of organic aerosols from Chinese residential cooking: case study in urban Guangzhou, south China, Journal of Atmospheric Chemistry, 72, 1-18, 10.1007/s10874-015-9298-0, 2015.